# Timing of exotic, far-travelled boulder emplacement and paleo-outburst flooding in the central Himalaya

Marius L. Huber[1,2], Maarten Lupker[1], Sean F. Gallen[3], Marcus Christl[4], Ananta P. Gajurel[5]

[1]Geological Institute, Department of Earth Sciences, ETH Zurich, Zurich 8092, Switzerland
[2]now at: Université de Lorraine, CNRS, CRPG, F-54000 Nancy,
[3]Department of Geosciences, Colorado State University, Fort Collins, Colorado 80523, USA
[4]Laboratory of Ion Beam Physics (LIP), Department of Physics, ETH Zurich, Zurich 8093, Switzerland
[5]Department of Geology, Tribhuvan University, Ghantaghar, Kathmandu, Nepal

*Correspondence to*: Marius L. Huber (marius.huber@univ-lorraine.fr)

**Abstract.** Large boulders, ca. 10 m in diameter or more, commonly linger in Himalayan river channels. In many cases, their lithology is consistent with source areas located >10 km upstream suggesting long transport distances. The mechanisms and timing of "exotic" boulder emplacement are poorly constrained, but their presence hints at processes that are relevant for landscape evolution and geohazard assessments in mountainous regions. We surveyed river reaches of the Trishuli and Sunkoshi, two trans-Himalayan rivers in central Nepal to improve understanding of the processes responsible for exotic boulder transport and the timing of emplacement. Boulder size and channel hydraulic geometry were used to constrain paleo-flood discharge, assuming turbulent, Newtonian fluid flow conditions, and boulder exposure ages were determined using cosmogenic nuclide exposure dating. Modelled discharges required for boulder transport, of ca. $10^3$ to $10^5$ m$^3$/s, exceed typical monsoonal floods in these river reaches. Exposure ages range between ca. 1.5 and 13.5 kyrs BP with clustering of ages around 4.5 – 5.5 kyrs BP in both studied valleys. This later period is coeval with a broader weakening of the Indian summer monsoon and glacial retreat after the Early Holocene Climatic Optimum (EHCO), suggesting Glacial Lake Outburst Floods (GLOFs) as a possible cause for boulder transport. We, therefore, propose that exceptional outburst events in the central Himalayan range could be modulated by climate and occur in the wake of transitions to drier climates leading to glacier retreat rather than during wetter periods. Furthermore, the old ages and prolonged preservation of these large boulders in or near the active channels shows that these infrequent events have long-lasting consequences on valley bottoms and channel morphology. Overall this study sheds light on the possible coupling between large-infrequent events and bedrock incision patterns in Himalayan rivers with broader implications for landscape evolution.

## 1 Introduction

Active tectonics, steep topography, dynamic surface processes, and extensive glacier cover expose the Himalaya to a range of catastrophic events that remain relatively rare on observational time scales and hence are poorly understood. Amongst the most

striking manifestations of catastrophic events are high magnitude earthquakes (Avouac, 2003) and resulting widespread landsliding or valley fills (e.g. Schwanghart et al., 2016a; Roback et al., 2018), and lake outburst floods (LOFs) whereby large volumes of impounded water is suddenly released into the fluvial network (Ives et al., 2010; Ruiz-Villanueva et al., 2017). LOF events in the Himalaya received widespread attention as the generated discharges may exceed typical precipitation-

35 induced floods by orders of magnitude (e.g. Costa and Schuster 1988; Cenderelli, 2000; O'Conner and Beebee, 2009; Korup and Tweed, 2007; Wohl, 2013; Cook et al., 2018). LOFs represent both a significant hazard (Kattelmann, 2003; Schwanghart et al., 2016b) and an active geomorphic agent of landscape evolution (Wohl, 2013; Cook et al., 2018; Turzewski et al., 2019).

LOF generation can be related to the formation of proglacial lakes at higher elevations, as glacier dynamics or frontal moraines

trap meltwater that, when rapidly released, can generate glacier lake outburst floods (GLOFs) (Richardson and Reynolds, 2000). LOFs may also be linked to the sudden damming of river channels by large landslides reaching the valley floor. The impounded water is then prone to catastrophic release as landslide lake outburst floods (LLOFs). A recent inventory of modern GLOF occurrences (1988 – 2017) along the entire Himalayan range shows a recurrence of ca. 1.3 significant GLOF events per year (Veh et al., 2019). A modern inventory of LLOFs has not been compiled as systematically, but recent reviews also suggest

widespread occurrences along the Himalayan range (Ruiz-Villanueva et al., 2017). Since most of the existing information about LOFs in the Himalaya is derived from observational or historical records, limited insight is available with respect to the maximum magnitudes that can be expected for these events or the evolution of their occurrence frequency through time, for example in response to climatic change.

The majority of LOF events originate in sparsely populated areas, but the steep slopes and high connectivity of upstream fluvial networks mean that flood waves can travel significant distances downstream with adverse effects for population and infrastructure (e.g. Gupta and Sah, 2007; Ziegler et al., 2014; Schwanghart et al., 2016b). The catastrophic draining of impounded lakes generally occurs rapidly, evoking spiky hydrographs (Cenderelli, 2000) and leaves little time for early warning, protection or evacuation measures. Better constraining the controls on LOF magnitude and frequency is, therefore,

imperative for improved risk assessments in the Himalaya, especially since these risks may evolve due to anthropogenic climate change and increasing land use change (e.g. Korup and Tweed, 2007; Huggel et al., 2012; Stoffel et al., 2014).

LOFs are generally infrequent but have the potential to provoke rapid incision or aggradation in fluvial systems with a long-lasting impact on erosion rates, sediment yields, and landscape morphology (e.g. Davies and Korup, 2010; Worni et al., 2012;

Lang et al., 2013). Tectonically active landscapes develop equilibrium and steady-state topography by balancing rock uplift and erosion over long timescales (>1 Myr) (e.g. Willett and Brandon, 2002). On shorter timescales ($<10^5$ yr), landscape evolution is characterized by phases of erosion and aggradation induced by changes in climatic forcing and the frequency of flood capable of breaching thresholds of bedload entrainment and bedrock detachment (e.g. Bull, 1991; DiBase and Whipple, 2011). LOFs have the potential to move coarse grain-sizes through the fluvial network that would otherwise be immobile

during typical floods. These events may hence promote rapid incision in the upper reaches of mountainous catchments where typical monsoonal discharges are too low to mobilize large grain sizes that cover channel beds, thereby setting the pace of landscape evolution (Cook et al., 2018). The timing, frequency and magnitude of these events, as well as their impact on landscape evolution as discontinuous, singular events is however still poorly understood and documented, emphasizing the need to bridge the gap between modern observations and the long-term evolution of landscapes.

To better understand catastrophic erosion and mass transport events as well as their potential impact on landscape evolution and related hazard, we focused on the occurrence, provenance, and mechanisms and timing of large boulder emplacement in central Himalayan river valleys. In the studied valleys, numerous large boulders of ca. 10 m in diameter and more, are found in or near the present-day channel beds. The lithology of many of these large boulders differs from those present on the adjacent

hillslopes but are the same as geologic units located 10's of kilometres upstream. Their elevation, well below the last and previous glacial maximum ice extents (e.g. Owen and Benn, 2005; Owen and Dorch, 2014; Owen, 2020), excludes glacial transport. The exact transport mechanisms of such exceptionally large grain sizes remain unknown and may be linked to reoccurring catastrophic events such as LOFs. In this contribution, we test this hypothesis using the boulder geometry and channel hydraulic geometry to estimate paleo-discharges required for their mobilization and use cosmogenic nuclide exposure

dating to constrain their emplacement age. We discuss our findings in the context of landscape evolution and natural hazards in the Himalaya and point to possible directions for future research.

## 2 Study area

The central Himalaya of Nepal accommodate ca. 18 mm/yr of the convergence between India and Eurasia (Ader et al., 2012) that translates into high tectonic uplift rates reaching 10 mm/yr (Bilham et al., 1997; Lavé and Avouac, 2001) and a dramatic

topographic gradient where elevations rise from about 200 meters above sea level at the range front to >8000 m of elevation across a horizontal distance of ~150 km. Great earthquakes (Mw ≥ 8.0) along the central Himalayan front happen irregularly with estimated reoccurrence intervals of less than 750 to 1000 years (e.g. Bollinger et al., 2014; Sapkota et al., 2013; Stevens and Avouac, 2016).

A main fault present on the central Himalayan mountain front is the Main Central Thrust (MCT), first named by Heim and Gansser (1938) (Figure 1). The MCT marks a distinct change in rock type and separates Lesser Himalayan rocks, which are mostly phyllites, slates, schists, metasandstones, limestones, and dolomites, as well as minor amounts of gneiss (e.g. the "Ulleri gneiss"), from crystalline Higher Himalayan rocks, which includes mostly augen-, ortho-, and para-gneisses (leucogranites and minor quartzites) within the central Himalayan region (e.g. Gansser, 1964; Stöcklin, 1980; Shrestha et al., 1986; Amatya

and Jnawali, 1994; Upreti, 1999; Rai, 2001; Dhital, 2015). Lower-grade metamorphic rocks of the Lesser Himalaya show prograde metamorphism related to MCT thrusting and increasing in metamorphic grade toward the MCT (from greenschist to

amphibolite in the vicinity of MCT). The higher Himalayan units have higher metamorphic grades (amphibolite to granulite facies) compared to the overridden Lesser Himalayan rocks (e.g. Pêcher, 1989; Rai, 2001).

The Trishuli and Sunkoshi/Balephi rivers are the targets of this investigation and are located in central Nepal, northwest and northeast of Kathmandu, respectively. They span the Greater Himalayan range from the arid high-elevation Tibetan Plateau to the Himalayan range front and are tributaries of the Narayani and Kosi Rivers, respectively. Both catchments are separated by a main drainage divide in their headwaters and the moderately-sized Melamchi Khola drainage basin at lower elevation (Fig. 1). The Indian summer monsoon, with highly seasonal distributed rainfall, affects both catchments with peak rainfall from

June to September (Bookhagen and Burbank, 2010) that is marked by high river discharge and sediment transport (Andermann et al., 2012).

GLOF events have been reported for both rivers. The Trishuli has been affected by the breach of lake Longda-Cho in Tibet during August 1964 (ICIMOD, 2011), but little is known about the downstream impact of that event. The Sunkoshi was

affected by GLOF events in 1935, 1964, 1981 and 2016 (Shresta et al., 2010; ICIMOD , 2011; Cook et al., 2018) and the last event of 2016 was closely monitored, revealing the impact of the GLOF, originating from the breach of a Tibetan proglacial lake, on the channel bed morphology (Cook et al., 2018). The upstream catchments of both river-reaches are heavily glaciated and a number of potentially hazardous lakes have been identified (Maharjan et al., 2018; Liu et al., 2020) so that it is likely that GLOF activity extended in the past.

**3 Material and methods**

**3.1 Sampling**

Fieldwork took place during two field campaigns in May and October/November of 2016 along the Trishuli and Sunkoshi main trunk river channels. In the field, boulders visibly larger than the surrounding overall bedload sediment grain-sizes were sampled for surface exposure dating and boulder provenance analysis (Figure 2, Table 1, Supplement, S1). Samples consisted

of ca. 1 to 2 kg of fresh rock from the top of the boulders. Fractured or weathered surfaces on boulders were avoided to minimize potential complications associated with post-depositional erosion. Boulders that exhibited clear evidence of post-depositional movement, for example, recent toppling, were not sampled. For large boulders that showed "pristine" surfaces and no evidence of post-depositional movement, flat surfaces were sampled in areas that minimized chances for burial by sediment or vegetation (e.g. the top of the boulder) (Figure 2). Rock material was removed using a chisel or a blade saw, and

the average sample thickness was recorded (S1 and S3). Topographic shielding was measured in the field with a laser range finder, neglecting vegetation.

## 3.2 Paleo-hydrologic discharge estimation

Paleo-discharges necessary to transport large grain sizes were computed based on boulder sizes and channel hydraulic geometry. Boulder diameter was determined from high-resolution satellite imagery (Google Earth) as field measurements were found impractical for boulders in the active channel. A succession of imagery of the same object covering several years was used to calculate an arithmetic mean for boulder diameter to minimize error due to imagery distortion (Table 1). We cross-checked our boulder size estimates by comparison with objects of known size both on high-resolution satellite imagery (Google Earth) and pictures taken during field campaign (e.g. persons, vehicles). The minimum diameter in bird's-eye view was taken as the intermediate diameter under the assumption that clasts were emplaced with approximatively vertical short axes. Rock density was estimated based on typical densities reported for the sampled lithologies. Hydraulic geometry measurements included valley cross-sections and channel bed slopes, which are needed to calculate flow channel characteristics during emplacement and were obtained by digitizing 1:25 000 and 1:50 000 scale maps of the Survey Department of Nepal (S2). These estimates were made using straight river reaches directly upstream of the studied boulders and were found to better render the channel cross-section shape compared to global digital elevation models we could access for this study.

Three different approaches were adopted to derive peak discharge values from the boulders surveyed in the Trishuli and Sunkoshi River valleys: i) the empirical approach after Costa (1983), based on a literature compilation describing average flow velocity that led to the transport of large boulders; ii) Clarke (1996)'s force balance approach which computes cross-sectional averaged velocities based on methods described by Costa (1983) and Bradley and Mears (1980); iii) the approach of Alexander and Cooker (2016) that allows flow velocities to be estimated from a theoretical force balance equation and taking into account an impulsive force to account for "inherently unsteady" and "non- uniform" flow. All three approaches take advantage of the incipient motion principle and compute velocities for turbulent, Newtonian fluid flow when a sediment particle of given intermediate particle diameter initiates motion on the stream's bed. The Gauckler-Manning formula (Gauckler, 1867; Manning, 1891) was then used to estimate peak discharge from the peak flow velocity determined above and the valley cross-sectional geometry using the numerical optimization scheme of Rosenwinkel et al. (2017). Peak discharges and maximum flood heights were calculated from flow velocities using the average of two valley cross-sections and bed slope for each boulder considered (S2). Access to Matlab scripts used for these calculations are detailed in the code availability section below.

## 3.3 Surface exposure dating

The samples for Beryllium-10 ($^{10}$Be) exposure dating were prepared in the laboratories of the Geological Institute in the Earth Science Department at ETH-Zurich. Bedrock was crushed and sieved to 250-1000 µm. Quartz was isolated by magnetic separation followed by five to seven sequential leaching steps with $H_2SiF_6$ and HCl (2/3 – 1/3 by volume). Meteoric beryllium (Be) was removed by three additional leachings with HF to dissolve ~10% of the sample mass at each step (Lupker et al. 2012). Pure quartz was dissolved after the addition of ca. 250 µg of $^9$Be carrier solution and Be was isolated using sequential ion

exchange column chromatography. [10]Be concentrations were measured for each sample at the Laboratory of Ion Beam Physics of ETH Zurich using the 0.5 MV TANDY AMS facility (Christl et al., 2013). Cosmogenic exposure ages were computed from the blank corrected [10]Be/[9]Be ratios using the online "Cosmic Ray Exposure program" (CREp) calculator (http://crep.crpg.cnrs-nancy.fr, Martin et al., 2017) using a global production rate $4.08 \pm 0.23$ at/g/yr, the ERA40 standard atmosphere and a correction for geomagnetic dipole moment changes. See Supplement (S3) for more detailed information on sample preparation and exposure age calculation.

## 4 Results

### 4.1 Field survey and boulder lithology

We surveyed and sampled a total of 16 boulders: ten boulders in the Trishuli River valley, six in the Sunkoshi River valley (Figure 1 and 3; Table 1). Along the Trishuli, the sampled boulders are found in different locations and configurations and have intermediate diameters ranging between 8.5 to 18.6 m. Detailed descriptions of the boulders are in S1. In the upstream part of the studied reach, near the village of Betrawati (N27.974; E85.184), boulders are located both in the modern floodplain close to the channel and on top of terrace deposits, ca. 18 m above the present-day channel elevation. Additional boulders ca. 15 km farther downstream, close to the village of Devighat (N27.859; E85.109), are deposited at the margin and close to the present-day channel on a tributary fan. Four of ten boulders in the Trishuli valley consist of gneiss, most likely orthogneiss originating from an intrusive protolith. Three gneiss boulders are located downstream of Devighat on a tributary fan and one gneiss boulder is found north of Betrawati (Figure 2, Table 1). In both locations, the surrounding hillslopes are composed of metasedimentary rocks of clearly differing fabric, so it is unlikely that these boulders are locally derived. The other boulders surveyed in the Trishuli valley are bluish or greenish phyllitic schists with a high phyllosilicate content with slightly differing fabrics among samples (Figure 2, Table 1). These boulders have a lithology compatible with the bedrock in the adjacent hillslopes, but also areas farther upstream.

Along the Sunkoshi, all the studied and sampled boulders were found within a ca. 3 km from the town of Balephi (N 27.732; E 85.780) and downstream along the Sunkoshi. The six studied boulders have intermediate diameters ranging from 4.5 to 29.9 m (Figure 2, Table 1). Only one boulder sampled and surveyed was not exposed in the river channel but was embedded in a terrace deposit just upstream of Balephi Khola's confluence with the Sunkoshi main trunk (Figure 2). The lithology of these Sunkoshi boulders consists of a variety of gneisses (Figure 2, Table 1); some show big porphyric felspar laths deformed to augen structures (NEQ/162 79, 80 and 98). Boulder sample NEQ/162 03, the largest boulder surveyed in this study, has an augen-fabric that looks more homogenous and slightly less deformed than the other samples (Figure 2, Table 1). The lithology of all boulders sampled in the Sunkoshi is different from the schists found in bedrock on the adjacent hillslopes, and they are thus, not locally derived but allochthonous.

## 4.2 Paleo-hydrologic discharge estimation

The average flow velocity values calculated after Costa's (1983) empirical approach are only dependent on the intermediate boulder diameter and range between 7.8 and 16.7 m/s for the surveyed boulders (S2). Flow velocities calculated using a fluid density of 1500 kg/m$^3$ range between 6.7 and 17.3 m/s using the Clarke (1996) method and between 4.3 and 11.2 m/s using Alexander and Cooker (2016) (S2). Costa's (1983) and Clarke's (1996) approaches produce similar results, Alexander and Cooker's (2016) method yields values less than an order of magnitude lower (Figure 3). Paleo-discharge derived from these velocities are shown in Figure 3 (and table in S2). Values of Costa (1983), Clarke (1996) and Alexander and Cooker (2016) derived peak discharges range from 3.63 x 10$^3$ to 1.97 x 10$^5$ m3/s for boulders in the Trishuli and 1.34 x 10$^3$ to 1.03 x 10$^5$ m$^3$/s for the boulders in the Sunkoshi.

## 4.3 Boulder exposure ages

[10]Be cosmogenic exposure ages of boulders in the Trishuli valley have ages ranging between 2.81 and 5.84 ka BP (Figure 4; Table 1) with overall 1σ uncertainties ranging from 0.35 to 0.67 ka and mode of the summed probability density functions around 4.5 kyrs BP. One of the ten boulders sampled upstream of Betrawati (NEQ/162 59; Figure 4; Table 1; S3), shows a significantly younger exposure age of 1.06 ± 0.29 ka BP. This outlier concentration probably reflects the effects of erosion or more likely burial of the sampled flat top-surface in the terrace deposit during a more recent stage in time and thus may not represent the true emplacement age.

The Sunkoshi exposure ages can be assigned to three different age groups (Figure 4): For one boulder, with the smallest intermediate diameter in this study of 4.5 m (NEQ/161 02), the low [10]Be concentration only allows to determine a maximum age of 0.49 ka BP (Figure 2 and 4; Table 1; S3), which could indicate recent movement (emplacement or toppling from an adjacent terrace deposit). The next three older boulders (NEQ/161 01, NEQ/162 98 and NEQ/162 79) show consistent ages within 1σ-uncertainty with 4.98 ± 0.65, 4.97 ± 0.51 and 6.23 ± 0.92 ka BP (Figure 2 and 4; Table 1; S3). Two older boulders are located in the Sunkoshi main channel with exposure ages of 10.96 ± 0.73 ka BP and 13.28 ± 0.96 ka BP (NEQ/162 80 and NEQ/162 03, respectively) (Figure 2 and 4; Table 1; S3).

## 5 Discussion

### 5.1 Boulder provenances and travel distances

The schist and phyllite boulder lithologies in the Trishuli valley are associated with the Paleoproterozoic lower Nuwakot group of the Lesser Himalayan sequence and more precisely from the Kuncha formation or the overlying Dangdagaon phyllites that could be found locally in the valley (Stöcklin, 1980; Upreti, 1999). The Trishuli phyllite and schist boulders might, therefore, originate from adjacent hillslopes. However, similar to the gneiss boulders, they are, in most cases, sub-angular to sub-rounded

with crescentic abrasion marks suggesting substantial fluvial transport distances (S1). Diagnostic mineralogy and fabric of the other surveyed ortho-gneiss boulders in the Trishuli catchment are not present in the Lesser Himalayan sequence and, therefore, must originate from areas upstream (or structurally above) of the MCT (Figure 1). No known Higher Himalayan unit klippe is mapped on hillslopes directly above the studied reaches, and the mixed boulder lithologies make a local emplacement source through mass-wasting unlikely. These observations, therefore, require minimum transport distances of approximately 22 km to 46 km depending on the present boulder location (Table 1).

A variety of gneiss boulder lithologies are found in the Sunkoshi/Balephi Khola catchment. Ortho-gneiss and augen-gneiss lithologies among boulders around Balephi are of higher metamorphic grade (amphibolite to granulite facies) and must originate upstream to the north across the MCT from Higher Himalayan crystalline rocks present in these areas or from gneiss to be found just below the MCT in the Lesser Himalayan footwall ("Ulleri type augen-gneiss" for NEQ/162 03, see S1) (Shrestha et al., 1986; Amatya and Jnawali, 1994; Dhital, 2015). In the absence of known Higher Himalayan units directly upslope above this reach of the Sunkoshi, this analysis suggests minimum transport distances of ca. 11 to 17 km (Figures 1, Table 1). While the surveyed boulders are mostly located below the Sunkoshi/Balephi Khola confluence and could, therefore, have been transported by both rivers, field observations show an abundance of boulders present in the bed of the Balephi Khola.

As noted above, boulders in both valleys are well below the extent of alpine glaciers in the modern or during the last and previous glacial maximum stages and their associated glacial deposits (e.g. Shiraiwa and Watanabe, 1991; Owen and Benn, 2005; Owen and Dorch, 2014; Owen, 2020; Figures 1 and 2). The low elevations where the exotic boulders are presently observed excludes a glacial transport mechanism. Rather, the observed locations and our provenance analysis indicate that the mobilization and transport of large grain-sizes occurred in central Himalayan river valleys over long distances (>10 km), most likely through fluvial processes.

## 5.2 Paleo-discharge estimates

The range of discharge estimates derived in this work is a first-order estimate and carries important assumptions. The sediment concentration of the flow directly influences transport capacity through flow density and flow mechanics (e.g. Pierson and Costa 1987). In hyper-concentrated flows with 40 to 70 wt. % sediment entrained, non-Newtonian, plastic fluid behaviour and laminar flow can arise due to the establishment of shear strength in the fluid material (e.g. Pierson and Costa 1987). However, if the amount of sediment entrainment remains at the lower end of this "hyper-concentrated" range, flow mechanics are still adequately approximated by Newtonian, turbulent flow of a "clear" waterflood (Costa, 1984; Pierson and Costa, 1987; Pierson, 2005; Wang et al., 2009; Hungr et al., 2014) as was assumed here. Nevertheless, our calculations do not apply for higher sediment load conditions, for example, conditions associated with debris flows. Other uncertainty arises from the extraction of valley cross-sectional profiles using topographic maps (see S2 for more details). Terrace flats and channel widths are only

crudely represented and do not account for past channel morphologies before and during the time of the floods. Since detailed riverbed morphology is required for the hydraulic discharge calculation, additional uncertainty arises from the resolution of the data used here and the necessity of using the modern channel geometry for these calculations. We hypothesize that these uncertainties are the main reason for the discrepancies between paleo-peak-discharge estimates for boulders from a similar age range that were presumably moved during a single event (Figure 3). First-order discharge estimates for boulder transport of surveyed clast sizes, therefore, broadly range from ca. $10^3$ to $10^5$ m$^3$/s.

These estimates are corroborated by observed boulder movement under known discharges (Figure 3A) reported in Xu (1988) and Cook et al. (2018) in the upper Sunkoshi (some 30 km upstream of Balephi). During the 2016 GLOF event, Cook et al. (2018) report the movement of a ca. 5.7 m diameter boulder for mean flow velocities between 8.2 and 6.8 m/s. An earlier study also reported the movement of a boulder of ca. 11.3 m in intermediate diameter for water flow velocities between 8.4 and 8.0 m/s during the 1981 GLOF in the same reach (Xu, 1988). Velocity and discharge estimates broadly agree with our estimates derived from the Costa's (1983), Clarke's (1996) and Alexander and Cooker's (2016) relations for boulder incipient motion. It is important to mention that peak discharges cannot be directly compared because of the various distance to potential source areas.

To place our results in the context of previous studies, we compare our discharge to those from the literature and historical records. Cenderelli and Wohl (2001) compared seasonal high flow floods (SHFFs) with discharges of recent GLOF events and they appeared to be at least one order of magnitude higher than monsoonal precipitation peak discharges in the central Himalayan Mount Everest region for reaches spanning many 10's of kilometres downstream of the breach locations. Peak discharges reaching $10^5$ m$^3$/s substantial distances downstream have been documented or suggested for a few historical events in the Himalaya mainly associated to LLOF such as the Great Indus flood of 1841 (Mason, 1929; Shroder et al., 1991), the great outburst in April 2000 in the Tibetan Yigong Zangbo River (Shang et al., 2003; Delaney and Evans, 2015; Turzewski et al., 2019), and the large LLOFs at Dadu River and Yalong River in the years 1786 and 1967 in Sichuan province, China (Dai et al., 2005; Runqiu, 2009). These events are, however, rarely observed even though there is sedimentological evidence that large-scale LLOF events happened regularly throughout the Holocene within the same catchments (e.g. Hewitt et al., 2011; Wasson et al. 2013). To our knowledge, GLOF discharge estimates of historically documented events in the Himalaya reach ca. $10^4$ m$^3$/s (e.g. Vuichard and Zimmermann, 1987; Hewitt, 1982; Xu, 1988; Yamada and Sharma, 1993; ICIMOD et al., 2011; Cook et al., 2018) with Holocene reconstructed discharge estimates that exceeded $10^5$ m$^3$/s, such as that reconstructed by Montgomery et al. (2004) for the Tsangpo River gorge outburst flood for locations >10 km downstream of the paleolake.

Hydrological stations from the Department of Hydrology and Meteorology, Government of Nepal allow comparison of our paleo-discharge estimates to measured discharges over the last decades. For the Trishuli reach, station number 447 (N27.97, E85.18) near the town of Betrawati, is located in between the two studied upstream and downstream boulder fields (Figure 1).

For the Sunkoshi boulders, two stations provide background hydrological information (Figure 1). First, station number 620 (N27.80, E85.77) on the Balephi Kola, about 8 km upstream of the most-upstream boulder and 9 km upstream from the confluence with the Sunkoshi main stem (without any major tributary confluences). Second, station number 610 (N27.79, E85.9) on the Sunkoshi at Barabise that is located about 14 km upstream of the Balephi Kola and Sunkoshi confluence. Comparison of the estimated paleo-discharges, which range between ca. $10^3$ to $10^5$ m³/s, with flow duration curves of the hydrological stations, shows that flows needed to mobilize the studied boulders generally exceed the largest flows on record (Figure 3 and 5). Discharge records may also include LOF events and, therefore, not only reflect monsoonal precipitation driven discharge. This finding suggests that typical monsoonal floods are unlikely to have the ability to move boulders of exceptionally large size (<10 m) and that LOFs are the most likely events responsible for large boulder displacement. The reconstruction of flood duration or initial lake size is, however, hampered by the multiple possible lake locations, unknown breach mechanisms, and the large uncertainties of our paleo-flood discharge estimates. Such a reconstruction is beyond the scope of this work and would require dedicated and computationally expensive fluid flow numerical models (e.g. Carling et al., 2010; Denlinger and O'Connell, 2010; Turzewski et al., 2019).

## 5.3 Timing and boulder emplacement mechanisms

The boulder population in the Trishuli valley displays a broad range of exposure ages from ca. 3 to 6 kyrs, with the notable exception of NEQ/162 59, which is significantly younger at ca. 1 kyr (Figure 4). The NEQ/162 59 boulder is still buried to a large extent, and its flat surface is only slightly elevated from the surrounding ground surface, suggesting that it was likely covered by sediment until recently. There is a systematic offset in age (up to 2 kyrs) between older exposure ages of boulders currently located on the fill terrace near the town of Betrawati and younger boulder ages in the channel or further downstream in the wider valley reach (Figure 4). In the Sunkoshi/Balephi Kola, three exposure age groups can be identified: i) the youngest, but also smallest boulder surveyed only yields a maximum age of ca. 500 years, given measurement uncertainties, ii) a group of three boulders with exposure ages ranging from ca. 3.5 and 6.5 kyrs and iii) two, late Pleistocene boulders, one of which is the largest surveyed boulder in our dataset. Based on these exposure ages and considerations presented in the previous sections, three main scenarios can be proposed for the emplacement of the surveyed boulders: (1) repeated boulder transport events, (2) excavation of boulders from large, older, fill deposits and (3) catastrophic events that resulted in long-range fluvial transport of the boulders. These scenarios are further discussed in the following paragraphs.

Slow transport of the boulders during repeated catastrophic events requires reoccurring exceptional discharges throughout the Holocene, as suggested by the flow velocities that are required to mobilize these boulders. Such a scenario is compatible with repeated GLOFs that frequently affect central Himalayan valleys (e.g. Veh et al., 2020). However, it is unlikely that such a mechanism is able to explain the distribution of exposure ages observed in this study. Repeated transport episodes will displace and rotate boulders, successively exposing different faces to dosing by cosmic rays. In such a case, boulder exposure ages would show a broad distribution of ages, reflecting repeating episodes of transport, stabilization, and exposure. Therefore, the

320 consistency of the age distributions makes it unlikely that the suite of sampled boulders was transported during multiple events. This interpretation is also supported by the observation of similar age distributions for boulders in valleys separated by a main drainage divide; the clustering of boulder ages in adjacent valley hints at a process capable of affecting flood distributions in two rivers around the same time. It should also be noted that there is a likely upper limit to the survival duration of boulders in a fluvial channel, as fluvial abrasion and comminution processes during transport (Attal and Lavé, 2009; Carling and Fan, 325 2020) or while at rest in the channel bed (Shobe et al., 2016; Glade et al., 2019) will ultimately reduce the size of the boulder until it can be exported by more frequent smaller flows.

Large valley-fill deposits are frequently observed in Himalayan valleys and span a wide range of ages (Lavé and Avouac, 2001; Pratt-Sitaula et al., 2004; Stolle et al., 2017). The evacuation and re-incision of such deposits could expose boulders that 330 were entrained and emplaced during the initial event and result in apparent exposure ages that are contemporaneous of the incision of the fill instead of the boulder transport. Remnants of such large fill deposits are not visible on the valley flanks in the direct vicinity of the studied locations. A large (>100 m thickness), presumably Pleistocene, fill terrace is nevertheless present farther downstream of the Trishuli boulder locations (N27.888; E85.141), near the town of Trishuli Bazar (Lavé and Avouac, 2001). However, the highly weathered state of this fill deposit material suggests that large boulders would unlikely 335 be preserved intact in such a warm and humid climate. A fill deposit re-incision scenario would require these processes to be synchronous across the drainage divide to explain the broadly similar age cluster at 5 kyrs observed in the Trishuli and Sunkoshi reaches. Re-incision would also need to occur rapidly to explain the relatively narrow age distribution of boulders sampled at different locations along the stream, as slow excavation since the Pleistocene would result in a broad range of ages. While not impossible, this process is more complex and nuanced than our preferred interpretation of mobilization and 340 emplacement during a single large flood event. However, as we noted earlier, the upstream Trishuli reach, boulders in the active channel are systematically younger (1-2 kyrs) compared to boulders located on the adjacent terrace. We interpret these younger ages as the result of the shielding of the boulders trapped in this thin fill deposit (< 20m thickness) before being exposed when the river re-incised the deposit (Figures 2 and 4; Table 1; S1). This re-incision had, however, to occur rapidly, or the age difference between in-channel and terrace-top would be larger.

The third possible mechanism of emplacement is the mobilisation of these boulders by catastrophic high discharge events, capable of mobilizing boulders over long distances. Such a scenario would explain the age distribution of boulders with the existence of at least one event in the Trishuli, and at least two events in the Sunkoshi/Balephi Kola reaches. Such catastrophic events are also likely to induce downstream landsliding by undercutting channel banks (Cook et al., 2018) and hence to entrain 350 boulders from different lithologies along its flow path while resulting in similar exposure ages across lithologies once deposited. It would also explain the observation, in the Sunkoshi/Balephi Kola reach, that the smallest boulder has the youngest exposure age and the largest the oldest, which is coherent with the fact that the boulders of older, larger events are not remobilized by subsequent smaller ones (also suggesting that our dataset is likely biased towards large events, as the smaller

events are likely erased). We, therefore, suggest that catastrophic long-range transport and rapid emplacement is the most likely scenario to explain our data.

In the Trishuli reach, the timing of a single catastrophic event is most accurately recorded by the ages of boulders on top of the small fill terrace in the upstream reach, and the downstream boulders with a similar age (Figure 4) as upstream boulders in the active channel have likely experienced shielding by the terrace fill sediments as discussed above. This yields an emplacement age estimate for the Trishuli of around 5.0 ± 0.3 ka BP (arithmetic mean with 1σ error of boulder samples NEQ/162 45, 46, 47, 66 and 67; Table 1). In the Sunkoshi/Balephi Kola reach, the oldest event is recorded by two boulders, NEQ/161 03 and NEQ/162 80, transported during the same event or series of events (within dating uncertainty) that occurred at ~11 to 13 ka (Figure 4; Table 1). It is followed by a second event recorded by the exposure age of three boulders with ages that agree within dating uncertainty at around 5.4 ± 0.7 ka BP (arithmetic mean with 1σ error of boulder samples NEQ/161 01, 79 and 98; Table 1). Finally, a single boulder (NEQ/161 02. Table 1) only resulted in the determination of maximum exposure age based on the low [10]Be concentration of ca. 0.5 kyrs. This is the smallest boulder surveyed and could have been mobilised during a smaller recent LOF or alternatively could have toppled over or was affected by unrecognized large surface erosion, which would both result in younger exposure ages.

The relatively large spread of exposure ages compared to other settings such as moraine boulders, for instance, can likely be attributed to the fluvial setting. Boulders can be affected by nuclide inheritance, if surfaces were exposed for significant durations prior to entrainment, which would bias the ages too old. Erosion of the boulder surface, a plausible process for boulders sitting in the channel of a mountainous stream, would bias the ages too young. But only significant erosion, such as the fracturing of a sizeable part of the boulder surface (which was avoided for sampling if recognized in the field) would affect the exposure ages (a steady-state weathering of the boulder surface of 10 mm/kyr would reduce the exposure age of a 5 kyrs old boulder by ca. 150 yr). Partial cover by sediments is also a plausible explanation for the scatter in exposure ages, and we invoke this effect for the younger ages of the boulders in the channel of the upstream reach of the Trishuli or boulder NEQ/162 59. However, since these processes (inheritance, erosion or partial sediment cover) are stochastic, it would be expected that if they dominated the signal, exposure age distributions would be more widespread then what was observed.

Trishuli and part of the Sunkoshi/Balephi Kola boulders show a clustering around 5 ka BP across a major drainage divide in the central Himalaya (Figures 1 and 4). At least two older ages in the Sunkoshi indicate flooding during the late Pleistocene (Figure 4); similar ages were not observed in the Trishuli valley (Figure 4). The events that emplaced these large-sized boulders, ca. 10 m in diameter or more, are rare since they represent the remnants of the last largest floods that were not remobilized by subsequent floods (Carling and Tinkler, 1998). The resolution of [10]Be exposure ages and uncertainty on the state of dated boulder surfaces does not definitely point toward a single event that affected both catchments. Boulder emplacement could potentially be the result of a series of events that occurred in a short time period of a few hundred years

within the range of dating uncertainty (Figure 4). However, the assumption that the boulder emplacements only record a flood of maximum magnitude (see above), the rarity of emplacement, and the absence of obvious stratification in the fill terraces (Figure 2; S1) point toward synchronous mid-Holocene emplacement events in each valley.

The timing of the Trishuli and Sunkhosi/Balephi Kola events found in this study is comparable to another large event responsible for the extensive fill terraces found in the central Himalayan Marsyandi river valley about 90 km to the west of the Trishuli valley (Pratt-Sitaula et al., 2004). The authors of that study found large terraces ("Middle terraces") composed of heterolithic conglomerates and boulders with lithologic evidence for transport distances of over 40 km. These fill terraces were interpreted as the result of a single massive earthquake-triggered landslide event that caused a catastrophic debris flow. Recalibrated radiocarbon ages of that infill date back to 4.6 to 5.1 ka BP (Yamanaka, 1982; ages recalculated with OxCal online calibrator, Bronk-Ramsay, 2013). The ages determined for the Trishuli and Sunkhosi/Balephi Kola boulders are however older compared to the exposure age of another studied large, far-travelled, boulder capping the Pokhara formation, further West in Nepal (ca. 1680 C.E.) (Fort, 1987; Schwanghart et al., 2016a). The emplacement of this later boulder was linked to a historical 1681 C.E. earthquake even though the magnitude and epicentre of this later event remain poorly constrained (Chaulagain et al., 2018).

**5.4 Triggers of Holocene catastrophic LOFs**

Our results demonstrate that high magnitude peak discharge ($10^3$ to $10^5$ m$^3$/s) by lake outburst events (LOFs) are most likely needed to explain the emplacement of large boulders in the Trishuli and Sunkoshi drainage catchments and that these events were clustered in time at around 5 kyrs in at least two distinct valleys (Figures 3, 4 and 5). To attribute these events to LLOFs or GLOFs requires evaluating our data in light of typical earthquake recurrence times and regional climate variability. Earthquakes and their associated co-seismic landslides provide a mechanism that could synchronously emplace large landslide dams in main valleys across water divides, exposing downstream reaches to LLOF events. Climate variability can directly affect precipitation patterns and intensity during the monsoon or indirectly affect glacial dynamics, through its modulation of glacier extent and proglacial lake volumes. Climate can, therefore, also be invoked as a potential trigger of large LOF events in multiple valleys during a short period of time, as observed in our data.

The timing of the boulder emplacement in the Trishuli and Sunkoshi/Balephi Kola suggests that these high magnitude flows last occurred ca. 5 kyrs ago. Older, larger boulders in the Sunkoshi/Balephi Kola suggest flows of even higher magnitude affecting trans-Himalayan valleys in the late-Pleistocene. While the dating precision available cannot strictly point towards a synchronous emplacement amongst the studied valleys (and the Marsyangdi, Pratt-Sitaula et al., 2004), major to great earthquakes ($\geq$ Mw 7.0 and 8.0 respectively) periodically rupture large parts of the Main Himalayan Thrust resulting in surface ground shaking over distance > 100s of km along the Himalayan range (e.g. Bilham, 2019). These earthquakes can trigger a large surface response with intense co-seismic landsliding, as was observed during the 2015 Mw 7.8 Gorkha earthquake

(Roback et al., 2018). Although the 2015 earthquake did not result in significant valley blocking due to landsliding that can result in large LLOFs, the total volume of co-seismic landsliding scales with earthquake magnitude (Marc et al., 2016) and hence LLOFs might be more likely following earthquakes with magnitudes larger than Gorkha. Great earthquakes ($\geq$ Mw 8.0) in the Himalaya susceptible to trigger large volumes of landsliding beyond what was observed during Gorkha have estimated recurrence times of 750 to 1000 years (Bollinger et al., 2014; Sapkota et al., 2013). If great earthquakes had a large likelihood to trigger widespread LLOFs (as would be required to explain the emplacement of boulders and valley fills in two to three valleys), we would expect the central Himalayan valleys to be strewn with boulders of younger and possibly more diverse ages owing to the geologically frequent recurrence interval of such events. Larger events, such as the possibility for $\geq$ Mw 9.0 earthquakes, with a suggested recurrence interval of >800 years along the entire Himalayan arc (Stevens and Avouac, 2016), could also trigger rare and large LLOFs. However, large hillslope failures and valley fills have been shown to occur for lower earthquake magnitudes as well (e.g. Schwanghart et al., 2016a), and other controls on landslide initiation are likely important factors as well, e.g. hillslope saturation (Lu and Godt, 2013). We, therefore, consider it unlikely that LLOFs are restricted to ($\geq$ Mw 9.0) events. Furthermore, the likelihood of an earthquake triggering LLOFs in two to three valleys during the same event is small. While earthquake-triggered LLOFs cannot be excluded (and have been proposed for other catastrophic valley fills; Schwanghart et al., 2016a), we do not favour this explanation for the emplacement of the boulders that are the focus of this investigation.

Climate change may be another LOF trigger. Given the large size of the studied catchments as well as the modern discharge record (Figure 5), it is unlikely that exceptional and localised extreme rainfall could occur synchronously in two large valleys. However, climate, through its modulation of glacier dynamics and the creation of proglacial lakes, could control the occurrence of GLOFs. Multiple terrestrial records for the monsoon-influenced Himalaya and the Indian subcontinent indicate a wet and strong monsoon phase during the Early Holocene Climatic Optimum (EHCO), followed by a dry phase between 5 ka to 4 ka BP. This dry phase is recorded amongst other proxy records, by a compilation of Indian monsoon records (Herzschuh, 2006); the transition to more arid conditions in central India that lead to a vegetation transition towards $C_4$ grasses as recorded in the Lonar Crater Lake sediments (Sarkar et al., 2015); changes in moisture sources as indicated by a drop in $\delta^{18}O$ of the Guliya ice cap on the Tibetan plateau (Thompson et al., 1997) (Figure 6). In response to the decrease in monsoonal precipitation, a number of preserved glacial landforms in the central Himalaya show a phase of glacial retreat around 5 ka BP (Abramowski et al. 2003; Finkel et al. 2003; Gayer et al. 2006; Schaefer et al. 2008; Pratt-Sitaula et al. 2011; Figure 6). This retreat has not been observed in all studied valleys, but glacial moraines dated to around 5 ka are reported in the Langtang valley in the upstream Trishuli catchment and in the Macha Khola, Manaslu massif ($^{10}$Be ages; Abramowski et al. 2003), in the Khumbu Himal (Everest region) with the Thuklha stage ($^{10}$Be ages; Finkel et al., 2003), as well as $^{3}$He and $^{10}$Be ages derived by Gayer et al. (2006) for moraines in the upper Mailun valley, Trishuli catchment suggesting a regional glacial response to aridification (Figure 6). Although the relation between GLOF frequency and ongoing climate change remains unclear (Harrison et al., 2018; Veh et al., 2019), past large phases of glacier retreat have been suggested to increase GLOF frequency (e.g. Walder and Costa,

1996; Clague and Evans, 2000; Wohl, 2013). Moraine dams form especially if fast glacial retreat follows earlier advances (Korup and Tweed, 2007). The documented phase of glacial retreat at ca. 5 ka BP, therefore, represents a possible triggering mechanism for the emplacement of the studied boulders across river drainage divides as it may have resulted in the more-wide spread occurrence of glacier lakes prone to GLOFs. Climatic forcing of GLOFs is, therefore, a suitable explanation for the emplacement of far-travelled fills in other valleys in the central Himalaya (e.g. Pratt-Sitaula et al., 2004) and agrees with the

provenance of a number of gneiss boulders originating from the glaciated Higher Himalayan crystalline. It should also be noted that abrupt climatic shifts such as the one following the EHCO occurred overall less frequently since the LGM compared to the recurrence time of great Earthquakes.

The older exposure ages derived in the Sunkoshi catchment, including the largest boulder surveyed in this study (NEQ/161 03

with 29.9 m intermediate diameter, Figure 2 and 4; Table 1), could be attributed to an LGM or post-LGM glacial retreat that lead to events preserved in the channel till today. If correct, our dataset reflects that long-term climate modulated LOFs can alter Himalayan Valleys on $10^3$ to $10^4$ yr timescales.

## 5.5 Implications for erosion and geohazards in the Himalaya

This study found evidence for high-magnitude discharge events in the form of outburst flooding in central Himalayan river

valleys. A record of exceptional flooding is preserved not only in the form of large boulders but also in the form of large alluvial fills in Himalayan valleys (Lavé and Avouac, 2001; Pratt-Sitaula et al., 2004; Stolle et al., 2017). As mentioned before (Wohl, 2013; Cook et al., 2018), LOFs may be responsible for channel incision and lateral erosion in the upstream reaches of mountainous rivers but may also lead to aggradation and valley fills in the lower reaches with long-lasting impacts on river morphology. Large grain-size boulders in channel beds also have the potential to affect long-term channel incision patterns

(e.g. Fort et al., 2010; Shobe et al., 2016; 2018). LOFs may, therefore, exert a strong control on the timing and locus of fluvial incision. In this study, we suggest that the emplacement of large grain-sizes in the two studied Himalayan valleys are related to large GLOF events that followed a time of regional glacial retreat after the EHCO and at the close of the Last Glacial Maximum. This interpretation argues for a climatic control on incision rates and sediment export in the Himalaya. Furthermore, it leads to the counter-intuitive notion that the erosional engine might be most efficient during, or at the onset of, drier climatic

periods as they represent periods of amplified occurrence of large, bedload mobilizing LOF events. Additionally, the fact that these boulders are preserved in the modern-day channel and adjacent fill deposits suggests that channel incision halted at these locations for periods of time subsequent to extreme outburst flooding. These extreme outburst events could, therefore, be key in regulating the episodic nature of fluvial incision (e.g. Pratt-Sitaula et al., 2004; Dortch et al., 2011) and have a long-lasting effect on sediment fluxes exported downstream. Further work on the frequency and magnitude of these events is required to

improve our understanding of their role in long-term channel morphology and incision. This future work should also address the relative importance of these events with respect to other catastrophic events in the Himalaya as we note that GLOFs are not the only mechanism that allow for long-distance boulder transport and valley aggradation. This is illustrated by the large

Pokhara valley fill deposit and boulder emplacement in central Nepal which was attributed to a series of earthquakes (Schwanghart et al., 2016a; Stolle et al., 2017) and had a long-term impact on valley morphology and sediment fluxes (Stolle et al., 2019).

In the face of anthropogenic climate change, increasing land use by population growth and ongoing investments into hydropower and road development in central Himalayan valleys, GLOF hazard and risk needs to be better quantified (e.g.ICIMOD, 2011; Schwanghart et al., 2016b, McAdoo et al., 2018). The recent acceleration of ice loss across the Himalayas (Maurer et al., 2019) has led to an increase in proglacial lake water volumes (Wang et al., 2015; Nie et al., 2017) but, surprisingly, has not so far lead to an observable increase in GLOF frequencies (Veh et al., 2019). How these trends will evolve in the near future remains unknown. In this study, we suggest the occurrence of large GLOF events during the phases of glacier retreat after the LGM and at the mid-Holocene climatic transition. The exact timing of these GLOF events in comparison to the climate forcing remains elusive, so as their precise origin in the landscape but it may forewarn possible future scenarios with significant threats to downstream populations and infrastructure. It should, however, be noted that the volume of water that can be stored in proglacial lakes in the upper reaches of the Himalayan rivers at present day could be significantly lower than volumes stored in proglacial lakes during glacial retreat from larger glacial extent. If correct, expected future GLOFs magnitudes could be lower than during the last deglaciation and the Holocene.

## 6 Summary and conclusion

We provide field observations of large boulders, ca. 10 m in diameter or more, that show lithologic evidence for travel distances of over 10s of kilometres in two central Himalayan valleys, the Sunkoshi and Trishuli. These boulders are well below the LGM ice extent and therefore require exceptional flows to explain the long travel distances. Using the boulder's estimated sizes and an assumption on flow density and regime, we estimated that discharges of $10^3$ to $10^5$ m$^3$/s are required to mobilize these grain sizes. Such discharges are, however, higher than typical monsoonal floods as constrained by hydrological data from nearby locations and therefore suggest that boulder emplacement occurred through high magnitude and catastrophic lake outburst floods.

Cosmogenic nuclide exposure dating ($^{10}$Be) shows emplacement ages of between 4.7 - 5.3 ka BP in the Trishuli valley and between 4.7 - 6.1 ka BP and 11 -13 ka BP in the Sunkoshi valley. Our data suggest that the younger events are correlated across water divides and possibly also with other deposits in the central Himalaya (Marsyandi valley, Pratt-Sitaula et al., 2004). The trigger for these LOF events remains difficult to constrain owing to the limits in dating precision that do not allow to clearly identify one single event across drainage divide or multiple events occurring within a short amount of time (ca. 1 kyr). Landslide lake outburst floods following a large seismic event and extensive co-seismic landsliding remains a possible explanation for boulder transport and emplacement; however, in this case, we argue against this interpretation because the

typical recurrence time for great earthquakes in the Himalaya is on the order of 1 kyrs, relative to the evidence of mid-Holocene and late Pleistocene LOFs in this study. In the case of co-seismic LLOF events, we would expect boulder deposits to have younger and/or more widespread ages. Alternatively, glacier lake outburst floods are also able to generate the required discharges. Terrestrial climatic records show that the Indian continent and the Himalaya experienced a significant aridification trend between 4 to 5 ka BP that follows a period of intense monsoon during the early Holocene climatic optimum. Glacier reconstructions show evidence for regional glacier retreat during that period as a response to a weaker monsoon. We, therefore, suggest that the observed boulders were most likely mobilized by GLOF events occurring during phases of glacial retreat.

These findings and interpretations have important implications regarding the dynamics of channel incision and erosion rates in the Himalaya. The role of these exceptional events as geomorphic agents of landscape evolution has recently been emphasized. We supplement these findings by showing that the studied central Himalayan river channels preserve traces of these large events that occurred 4 to 14 kyrs ago, suggesting a long-lasting impact on sediment dynamics and channel evolution. Furthermore, even though there is currently no evidence for an increase in GLOF frequencies as a response to anthropogenic climate change in the Himalayas, the type of evidence put forward in this work suggests that major phases of glacier retreat may be associated with GLOFs that may, if not be more frequent than in more stable glacial conditions, at least be of larger magnitude.

# 7 Acknowledgments

We thank Georgina Bennett, two anonymous reviewers and Paul Carling for comments and suggestions that greatly helped clarify our contribution. Wolfgang Schwanghart is also warmly thanked for efficient editorial handling and comments on the different versions of this manuscript. This work was supported by the ETH research grant 15 15-2 and the Alumni-fonds associated with the Geological Institute, Department of Earth Sciences, ETH Zurich. Special thanks to Katie Schide, Shyam Khatiwada, Lena Märki and Akash Acharya for help in the field. Negar Haghipour is thanked for help with the cosmogenic nuclide sample preparation.

# 8 Code availability

All paleo-hydrologic discharge calculations can be repeated using a user-friendly Matlab script accessible via the URL https://gitlab.com/mlh300/bouldersforpaleohydrology/ and detailed in the affiliated explanatory file.

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

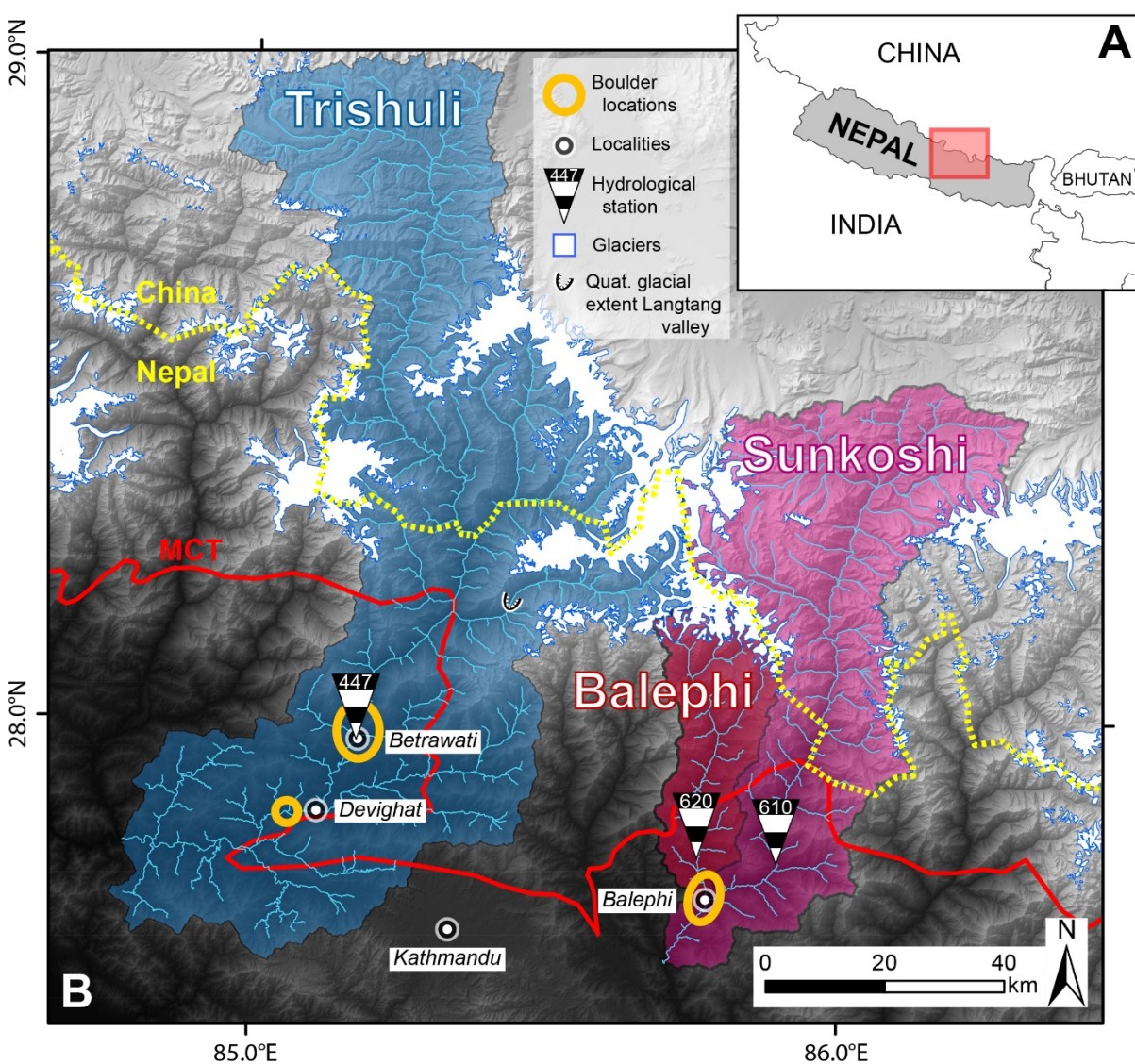

**Figure 1: A. Regional overview of the study area (red box). B. Location of the studied river reaches in central Nepal (main catchments of Trishuli and Sunkoshi, as well as Sunkoshi sub-catchment of Balephi), studied boulder locations and hydrological stations from the Nepal Department of Hydrology and Meteorology (along with station number). Also shown is the trace of the Main Frontal Thrust (MCT, see references in text), glacier cover (Global Land Ice Measurements from Space(GLIMS), retrieved March 2018) and an estimate of maximum Quaternary glacial extent in the Langtang valley by evidence of valley morphology (Shiraiwa and Watanabe, 1991). Shaded relief is based on digital elevation model AW3D30, © Japan Aerospace Exploration Agency.**

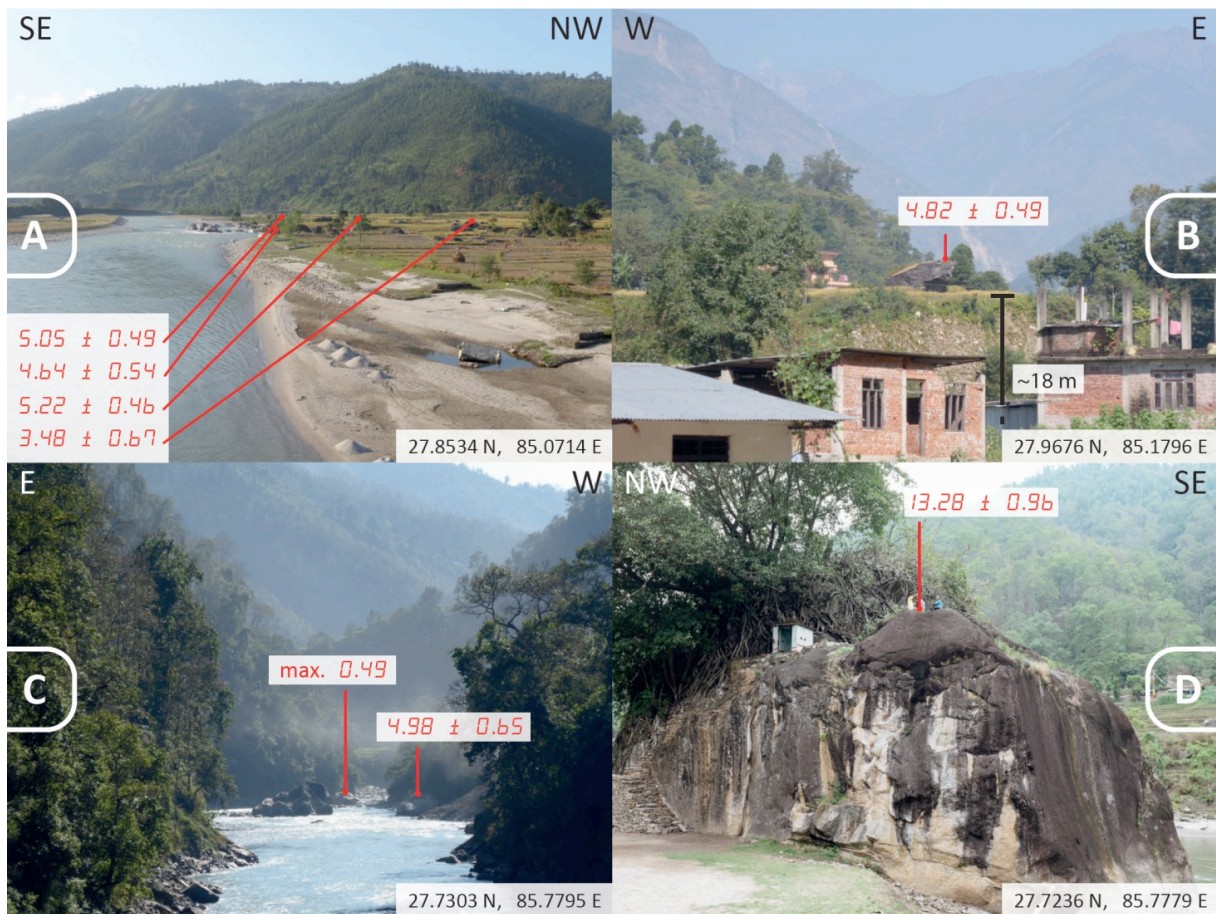

**Figure 2: A-** Boulders lying on a tributary fan south of Devighat, Trishuli valley. The main valley widens substantially at this location. Sample from top to bottom (as shown in the Figure): NEQ/162 47, ...46, ...45, ...44. **B-** Boulder appears sub-angular sitting on top of terrace deposit at Betrawatti (NEQ/162 66), Trishuli valley. In the background peaks rise >5000 m. **C-** Narrow part of Sunkoshi river after its confluence with Balephi Khola. Sample from top to bottom (as shown in the Figure): NEQ/161 02, ...01. **D-** Largest boulder surveyed in this study (NEQ/161 03) in Sunkoshi valley with an intermediate diameter of 29.9 m consists of gneiss lithology (Ulleri Gneiss), minimum travel distance 13 km. [10]Be surface exposure ages in ka BP. Coordinates of viewpoints given.

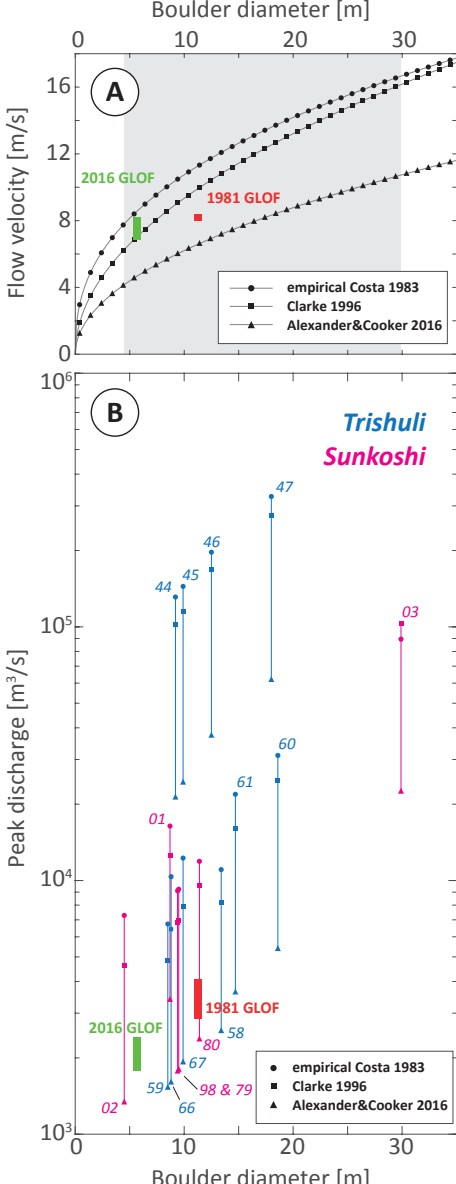

**Figure 3: A-** Theoretical flow velocities required to move boulders of a given diameter with explanatory input parameters (channel bed slope 0.03, rock material density 2700 kg/m3, fluid density 1500 kg/m3) according to the parametrisations and models of Costa (1983), Clarke (1996) and Alexander and Cooker (2016). The grey shaded area indicates range of boulder intermediate diameters from this study. Green and red rectangles are bounded by velocity estimates upstream and downstream boulders (5.7 m and 11.3 m in diameter) mobilised during the 2016 and 1981 GLOF events in the upper Sunkoshi (Cook et al., 2018; Xu, 1988). The Clarke (1996) method is plotted with a channel bed slope adjusted to Cook et al. (2018) which is the gradient of the Sunkoshi reach at the location of boulder movement (0.0245). **B-** Estimated peak-paleo-discharges required to move the studied boulders according to the three models that were used for paleo-discharge calculations. Green and red rectangles are bounded by upstream and downstream estimates of observed boulder movements for the 2016 and 1981 GLOF events in the upper Sunkoshi (Cook et al., 2018; Xu, 1988).

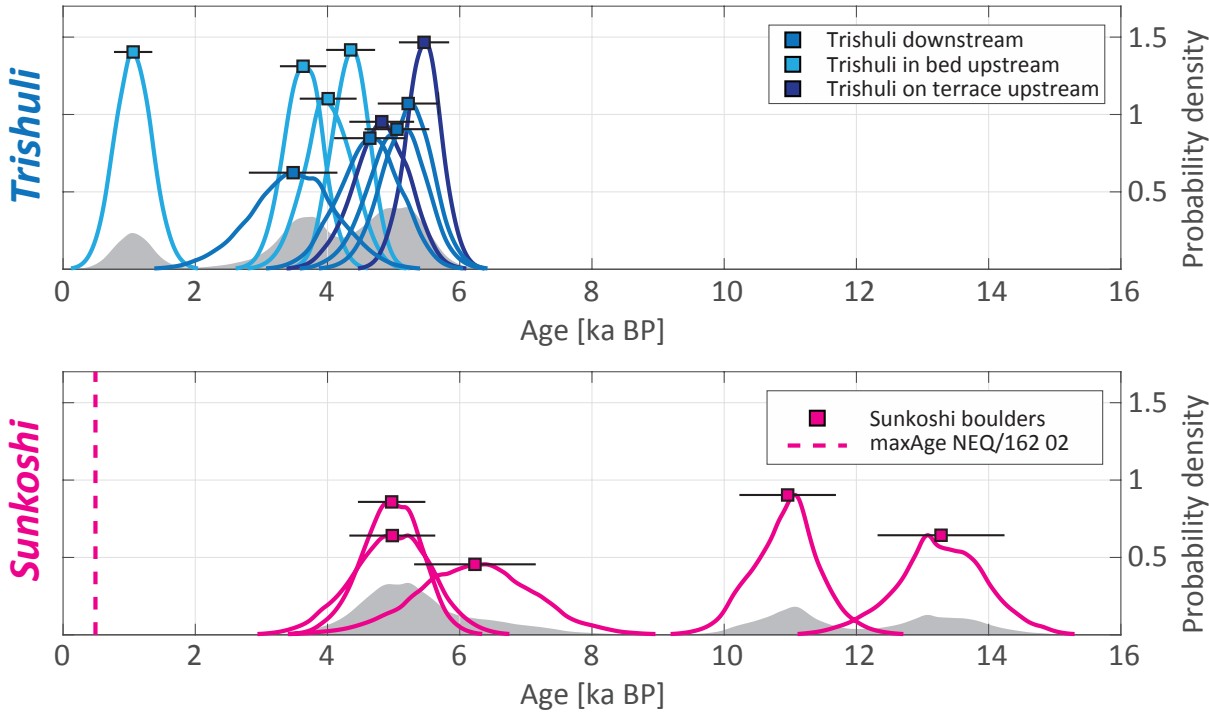

**Figure 4: Probability density function for the [10]Be boulder exposure ages in the Trishuli and Sunkoshi river reaches. The Trishuli boulders were subdivided in groups depending on boulder geographical location (upstream vs downstream) and position relative to the present-day channel. The grey shaded area in the background of each plot shows the cumulative sum of the probability distributions normalised by the quantity of boulders measured in the respective valley. Ages are indicated with 1σ error bars and were calculated using the online CREp calculator (Martin et al., 2017) accessed in June 2018 with a Sea Level High Latitude production rate of rate 4.08 ± 0.23 at/g/yr.**

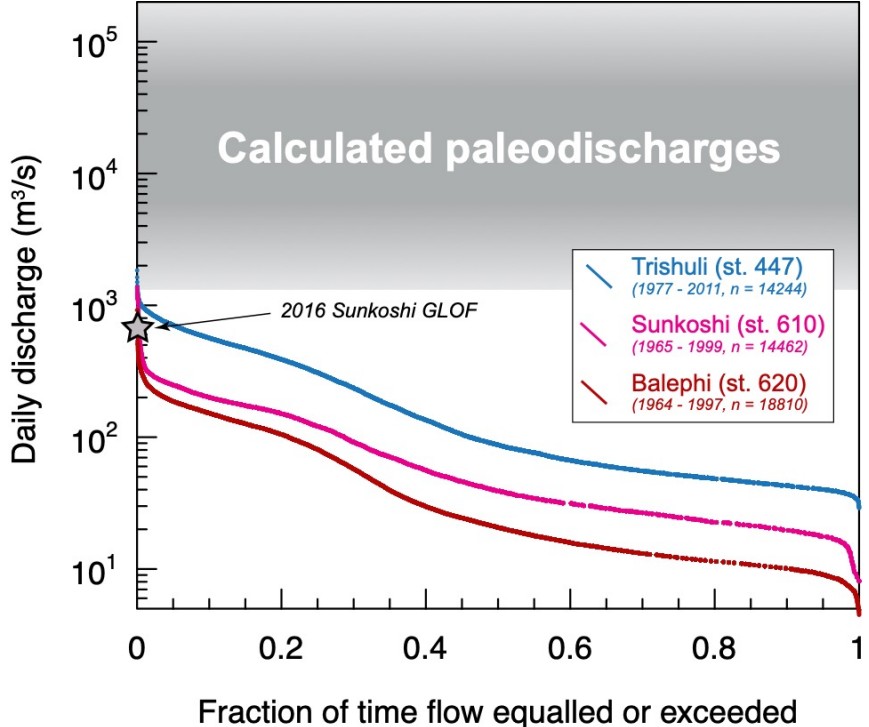

**Figure 5:** Flow frequency curves of daily discharge data from the Trishuli at Betrawati (Station n# 447), the Sunkoshi in Barabise (Station n#610) and the Balephi Khola in Jalbire (Station n#620) along with the period covered by the data and the number of daily discharge measurements (Data from the Government of Nepal, Department of Hydrology and Meteorology). The range of paleo-discharges required to mobilise the studied boulders is shown in shades of grey. The discharge measured in Barabise for the 2016 GLOF on the Sunkoshi is also shown for comparison (Cook et al., 2018).

835

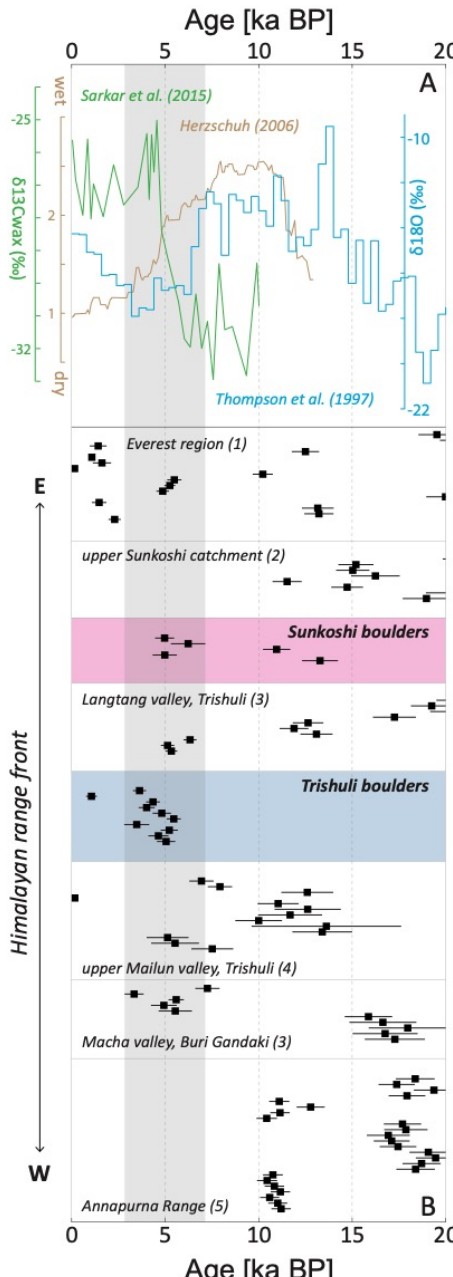

**Figure 6: A-** Climate proxy data: $\delta^{18}O$ ice core measurements recovered on the Qinghai-Tibetan Plateau, China by Thompson et al. (1997); mean effective moisture index compiled from multiple paleoclimatic records in Indian monsoon dominated Asia by Herzschuh (2006); $\delta^{13}Cwax$ (‰) vs VPDB lipid biomarker record from Lonar Lake central India by Sarkar et al. (2015). These proxies suggest a major climatic shift affecting monsoonal precipitation and consequently glacial dynamics on the south facing front at around 5 ka BP in the central Himalaya. **B-:** Boulder exposure ages in alignment with moraine deposits dated in the central Himalayan study region and recalculated consistently for better comparison: (1) Finkel et al. (2003), (2) Schaefer et al. (2008), (3) Abramowski et al. (2003), (4) Gayer et al. (2006), (5) Pratt-Sitaula et al. (2011). $1\sigma$ uncertainties are given with horizontal bars. Realm of boulder ages around 5 ka BP is accentuated with grey shading.

| sample # | catchment | Lat [°] | Lon [°] | Alt. [m a.s.l.] | boulder intermediate diameter [m] | Lithology | Age(ka) | 1σ (ka) | Minimum travel distance [km] (1) |
|---|---|---|---|---|---|---|---|---|---|
| NEQ/161 01 | SUNKOSHI | 27.72911 | 85.77910 | 674 | 8.7 | orthogneiss | 4.98 | 0.65 | 17 |
| NEQ/161 02 | SUNKOSHI | 27.72805 | 85.77883 | 672 | 4.5 | whitish orthogneiss | max. 0.49 | | 17 |
| NEQ/161 03 | SUNKOSHI | 27.72371 | 85.77810 | 668 | 29.9 | ortho-, augengneiss, Ulleri Gneiss | 13.28 | 0.96 | 13 |
| NEQ/162 44 | TRISHULI | 27.85610 | 85.06961 | 441 | 9.2 | orthogneiss | 3.48 | 0.67 | 46 (13) |
| NEQ/162 45 | TRISHULI | 27.85601 | 85.06905 | 440 | 9.9 | orthogneiss | 5.22 | 0.46 | 46 (13) |
| NEQ/162 46 | TRISHULI | 27.85551 | 85.06886 | 445 | 12.5 | orthogneiss | 4.64 | 0.54 | 46 (13) |
| NEQ/162 47 | TRISHULI | 27.85589 | 85.06848 | 445 | 18 | phyllitic schist | 5.05 | 0.49 | 46 (13) |
| NEQ/162 58 | TRISHULI | 28.00898 | 85.18383 | 679 | 13.4 | phyllite | 3.63 | 0.35 | 0 |
| NEQ/162 59 | TRISHULI | 28.00888 | 85.18438 | 680 | 8.5 | orthogneiss | 1.06 | 0.29 | 22 |
| NEQ/162 60 | TRISHULI | 27.96964 | 85.18269 | 593 | 18.6 | schist | 4.35 | 0.37 | 0 |
| NEQ/162 61 | TRISHULI | 27.96942 | 85.18208 | 593 | 14.7 | schist | 4.01 | 0.43 | 0 |
| NEQ/162 66 | TRISHULI | 27.97021 | 85.17987 | 613 | 8.8 | schist | 4.82 | 0.49 | 0 |
| NEQ/162 67 | TRISHULI | 27.97065 | 85.17986 | 613 | 9.9 | phyllitic schist | 5.46 | 0.38 | 0 |
| NEQ/162 79 | SUNKOSHI, Balephi | 27.73503 | 85.78021 | 680 | 9.5 | ortho-, augengneiss | 6.23 | 0.92 | 16 |
| NEQ/162 80 | SUNKOSHI | 27.73389 | 85.78328 | 695 | 11.4 | ortho-, augengneiss | 10.96 | 0.73 | 12 |
| NEQ/162 98 | SUNKOSHI, Balephi | 27.74063 | 85.77722 | 693 | 9.4 | ortho-, augengneiss | 4.97 | 0.51 | 11 |

(1) in brackets if drainage from eastern, non-glaciated and smaller tributaries is included (i.e. shortest distance to MCT)

**Table 1: Summary table of boulder location, lithology, minimum travel distance and exposure age. Further details can be found in the Supplement (S1 and S3).**