# Peer review of "Timing of exotic, far-travelled boulder emplacement and paleooutburst flooding in the central Himalaya"

_Earth Surface Dynamics, 2020_

## Short Comment (SC1) · 20 Mar 2020

For the supplement's title a correction is needed: "far-travelled" instead of "far-traveled", so the spelling is identical with the manuscript's title.
* * *

---

## Short Comment (SC2) · 25 Mar 2020

Perspective

Huber et al (2020) present a potentially valuable contribution to determining the rate and timing of transport of large boulders in a montane environment in the context of landscape evolution. However, herein I propose that a more cautious and nuanced approach might enhance the interpretations advanced. The initial motion and transport of large boulders in fluvial environments is poorly understood, but is an important topic of research related to channel and valley evolution. Indeed, the corollary is that identification of exceptionally large boulders that have not moved is also important as these

indicate an upper limit to palaeoflood competence (Carling & Tinkler, 1998; Carling et al., 2002).

Palaeohydraulics using boulder data

As noted by the authors, large boulders can be delivered to the valley bases by earthquakes and other gravitational slope processes. These slope-derived boulders have the potential to be transported by monsoon floods or by lake-outburst floods. Some rockfall blocks may remain in situ until abrasion and weathering reduces their size sufficiently for fluvial transport. Nonetheless, in the present examples, the boulders are many kilometres downstream from known outcrops of the given lithologies, so they must have been transported by ice or by water. Huber et al (2020) argue that the boulders must have been water-transported to their present locations as they are outside of the known last glacial ice-limits. Whilst this is a reasonable assertion, it is also possible that some of the boulders could have been transported by ice to moraines upstream (e.g., LIA, LGM) and then transported from the moraines to their current locations by fluvial processes. In addition, consideration needs to be given to the possibility that the boulders were transported by an early and more extensive glaciation, but the authors provide no information on the ice-limit of glaciations prior to the last. In the circumstances, the distances of travel induced by floods reported by the authors require recalculation or qualification. It would be useful to have the positions of the glacial maxima, if known, indicated on Fig. 1.

I assume boulder sizes were also measured in the field but the text suggests all dimensions were obtained from Google Earth imagery. There is no way to know the resolution of Google Earth imagery at a given site, which can range from 15m to 0.15m. Calibration of pixel resolution against objects of know size in the field would have been useful, as would publication of the dates of images used. Imagery also can provide only the planview size of boulders with the vertical dimension having to be estimated. Determining boulder volume in such circumstances is a well-known problem and use of laser-scanning would have produced more reliable boulder dimensions. That being

said, the limitations of the Huber et al. sampling methodology are not unusual and I do not wish to dwell unnecessarily on the sampling of boulder dimensions, but prefer to focus on the entrainment calculations and estimates of palaeodischarges, as well as the dating issue.

It would have been useful to see a fuller appreciation of the three entrainment equations utilized, with an explanation for their selection. The Costa (1983) empirical method for boulders in shallow flows has stood the test of time as it is equivalent to applying the Shields equation with an average value of the Shields parameter (Ïť) of 0.04 (s.d. of 0.011 for n = 40) for gravel in the size range 20mm ≤ D ≤ 5000mm (Carling & Fan, 2020). The use of this equation is an extrapolation in the analysis of Huber et al. (2020). Theoretical considerations have concluded that a Shields parameter of 0.04-0.05 applies to well-embedded large gravel clasts, including boulders (Lamb et al. 2015; van Rjin, 2019; Dey & Ali, 2019), but large boulders may be entrained for Ïť-values < 0.04; down to 0.01 for example (Carling & Fan, 2020). Protruding boulders in particular tend to have lower Ïť-values. In addition, boulders protruding from the flow and subject to obstacle-induced standing waves, have radically different entrainment conditions in contrast to fully-submerged boulders (Carling et al., 2002a & b). Scour in gravel surrounding boulders is far more intense for large protruding boulders than for deeply-submerged boulders (Schlömer et al., 2020) promoting undermining and motion of large protruding boulders (Clark, 1996). Boulder protrusion and undermining might be a particular issue, where there is a 'dramatic topographic gradient' as indicated by the authors resulting in steep channels subject to rapidly changing periods of aggradation and degradation of alluvial bed levels, also noted by Huber et al. (2020).

Clark (1996) used a simple force-balance for fully-protruding clasts with a drag coefficient of around 1.18 or less. This selection of drag coefficient values can only apply to fully-submerged boulders in deeper flows (Carling et al., 2002b). This appreciation explains why the Clarke function plots below the Costa function in Fig. 3A. Nevertheless, Clarke was concerned that some of his boulders may have been transported by

debris-flows, in which case the entrainment threshold calculated for clear-water using his equation might readily be related to observed boulders that are too large to have been moved by water. Consequently, extreme caution is needed when applying the Clarke analysis.

The Alexander & Cooker (2016) analysis was most welcome as it incorporates the small impulse factor due to floods arriving as a bore front, as well as flow instability. It is well-known in coastal science that very large shoreline boulders have been observed to move during relatively small storm-wave impacts that exert a large impulse. In these cases the application of the traditional force-balance approach for steady flow is not suitable and rarely predicts boulder entrainment accurately. Nonetheless, the Alexander & Cooker analysis applied to the data of Huber et al. (2020) lies below that of Clarke (1996) in Fig. 3A, as it should. Given the issue of imperceptible downstream creep of large boulders due to undermining, it is possible that the boulders observed were moved down valley under conditions whereby small values of Ïť pertain, or the boulders could have been emplaced by debris-flows which Huber et al. (2020) do not consider.

The arguments above explain why the velocity ranges estimated subsequent to application of the three different entrainment equations differ systematically. Taken together the velocities reported by Huber et al. (2020) range between 4.3msˆ-1 and 17.3msˆ-1. However, velocities in alluvial rivers rarely exceed 3msˆ-1 (e.g. Jia et al., 2016), with flow in steep bedrock gorges occasionally reaching 8 to 10ms-1 (Barnes, 1956; Pielou, 1998; Whipple et al., 2000). Natural dam failures can result in short-duration, modelled high-velocities (15-19msˆ-1: Alho et al., 2005; Carrivick, 2007; Carrivick et al., 2013) in upstream reaches which tend to reduce downstream as the floodwaves attenuate. Thus, although Huber et al. (2020) in section 2.3 provide some caveats as to the discharge estimates they regard discharges up to 10ˆ5 mˆ3sˆ-1 as acceptable. From the arguments I outline above the discharges at the higher end of the range calculated by Huber et al. (2020) should be viewed with extreme caution. In

making this assertion, I are not suggesting that floods of that magnitude have never occurred due to dam-breaks events. Indeed, the comparison of discharges advanced by Huber et al. (2020) with other large floods (c. 105 m3s-1) reported in the literature does indicate some precedence, but these earlier outbreak flood estimates, of course, are subject to considerable uncertainty. Despite relatively low velocities occurring during monsoonal floods in contrast to outbreak floods, the long time-periods (basically post-glacial) considered means that over thousands of years boulder creep becomes an important transport mechanism in steep mountain valleys, as well as debris-flow. In summary, it remains an open question as to whether large boulders can be transported by monsoon floods or whether exceptional outbreak floods necessarily have to be invoked.

Cosmogenic dating

The application of cosmogenic nuclides to date the deposition of the boulders raises a few issues that should prompt further consideration by the authors. There are four main points.

1) In the absence of burial, exposure ages should always be regarded as minimum ages.

Unless there are some preserved surface forms on these boulders, such as glacial striations, then one should assume that some fraction of the nuclides that arrived with each boulder have been lost via erosion. At what rate? It would be good to see a simple sensitivity analyses applying plausible surface erosion rates for this environment: 1 mm/kyr, 2 mm/kyr, 5mm/kyr, 10 mm/kyr? etc. The question is: what surface rate of erosion is required to move outside of the mid-Holocene window? How plausible is that?

2) Intermediate storage and accumulation of nuclides between source and sink.

Some of these boulders are big: Trishuli valley, ten boulders ranging 8.5 to 18.6 m;

Sunkoshi valley, six boulders ranging 4.5 to 29.9 m. These exhibit very long boulder transport distances: 22-46 km in Trishuli valley, and 11-17 km in Sunkoshi valley. Boulders that exhibited signs of post-depositional movement were not sampled; however, it is the transport history of these boulders that could be the problem here. It would be good to see some explicit acknowledgement of the possibility that some fraction of these boulders accumulated nuclides at some point upstream of where they were sampled. As noted above, these are very long transport distances. To expect a single transport from source to sink is simply not plausible for blocks of this size. It is more likely that these blocks have taken a few rest stops along the way where they might have been buried or eroded in situ, if close to the channel.

A simple test could have been to sample the underside of a few of the tabular blocks to determine whether they spent much time the other way up (e.g. Fujioka et al. 2015). Intermediate storage might, for example, be an alternative explanation for the two blocks in Sunkoshi with high nuclide abundances. It may be possible that boulder erosion during transport may have removed any inherited nuclides accumulated during intermediate storage steps. However, it would be good to see some consideration of this scenario in the manuscript. The authors do not air any of these potential doubts; I suggest a more candid discussion would be welcome.

3) Age clusters

Regarding the Sunkoshi data shown in Fig. 4, how is it that a single exposure age forms an 'age group'. These are single data points. The clustering of exposure ages around 4.5-5 ka is not convincing in Sunkoshi given the small number of samples, which is too few to establish any statistical significance. The Trishuli cluster is more convincing. One important thing that has been learned about Himalayan valleys over the past decade is that valley floor elevations fluctuate by tens or even hundreds of metres over relatively short intervals (e.g. Schwanghart et al. 2015; Munack et al. 2016). Boulders sampled up to 18 m above the channel in Trishuli valley suggests that there might be scope here to bury clasts for significant periods within the valley fills.

Perhaps an alternative explanation of your age distributions is that these nuclide concentrations reflect 'exhumation'. That is, the nuclides accumulated in the boulder surfaces as they were exhumed from beneath a deeper thickness of valley fill. The exhumation itself might be a pervasive erosion of the valley floors resulting from the postglacial decline in sediment supply. Such a scenario would yield the kind of spread with vague clustering. It would be interesting to investigate this further. Exhumation might explain why there are no boulders with higher nuclide abundances. If these boulders are the product of outburst floods, why do we have such a short record in these valleys? Where are the boulders deposited by previous outburst? All washed away?

4) Lower nuclide concentrations, why not younger floods?

In Section 4.3 it would be better to simply have the ages reported, rather than the special conditions under which some data are preferred over others. It is a bit unclear why samples with low nuclide abundances are dismissed as somehow problematic when they might just easily be the result of more recent floods?

Specific issues: In line 201, I do not follow why this low concentration 'only allows to determine a maximum exposure age of 0.49 ka'? Finally, I am a bit unclear on how the probability distributions for individual ages are calculated in Fig. 4 and what this means exactly. Perhaps some more detail is required here.

References

Alho P, Russell AJ, Carravick JL, Köyhkö J. (2005) Reconstruction of the largest Holocene jökulhlaup within Jökulsá á Fjöllum, NE Iceland. Quaternary Science Reviews, 24: 2319–2334.

Barnes HL. (1956) Cavitation as a geological agent. American Journal of Science, 254, 493–505.

Carling, P.A., Fan, X. (2020) Particle comminution defines megaflood and superflood energetics. Earth-Science Reviews, 204, 103087

Carling, P.A., Hoffmann, M., Blatter, A.S. (2002a). Initial motion of boulders in bedrock channels, in Ancient Floods, Modern Hazards: Principles and Applications of Paleoflood Hydrology, Water Science and Applications, 5, P. K. House, R. H. Webb, V. R. Baker, D. R. Levish, Eds. (American Geophysical Union), pp. 147-160.

Carling, P.A., Tinkler, K.J. (1998) Conditions for entrainment of cuboid boulders in bedrock streams: an historical review of literature with respect to recent investigations" in Rivers over Rock: Fluvial Processes in bedrock Channels, K.J. Tinkler, E.E. Wohl, Eds. (American Geophysical Union), pp. 19-34.

Carling, P.A., Hoffmann, M., Blatter, A.S. and Dittrich, A. (2002b) Drag of emergent and submerged regular obstacles in turbulent flow above bedrock surface. In: Rock Scour due to Falling High-Velocity Jets Editors A.J. Schleiss and E. Bollaert, Swets and Zeitlinger/Balkema, Lausanne.

Carrivick JL. (2007) Hydrodynamics and geomorphic work of jökulhlaups (glacial outburst floods) from Kverkfjöll volcano, Iceland. Hydrological Processes, 21, 725–740.

Carrivick JL, Turner AG, Russell AJ, Ingeman-Nielsen T, Yde JC. (2013) Outburst flood evolution at Russell Glacier, western Greenland: effects of a bedrock channel cascade with intermediary lakes. Quaternary Science Reviews, 67, 39–58.

Dey, S., Ali, S.K.Z. (2019) Bed sediment entrainment by streamflow: State of the science. Sedimentology, 66, 1449-1485.

Fujioka et al. (2015) Flood-flipped boulders: In-situ cosmogenic nuclide modeling of flood deposits in the monsoon tropics of Australia. Geology, 43, 43–46.

Jia Y, Wang Z, Zheng X, Li Y. (2016) A study on limit velocity and its mechanism and implications for alluvial rivers. International Journal of Sediment Research, 31, 205–211.

Lamb, M.P., Finnegan, N.J., Scheingross, J.S., Sklar, L.S. (2015) New insights into the mechanics of fluvial bedrock erosion through flume experiments and theory. Geomorphology, 244, 33-55.

Munack, H., Blöthe, J.H., Fülöp, R.H., Codilean, A.T., Fink, D., Korup, O. (2016) Recycling of Pleistocene valley fills dominates 135 ka of sediment flux, upper Indus River. Quaternary Science Reviews, 149, 122-134.

Pielou EC. (1998) Fresh Water. University of Chicago Press: Chicago, IL.

Schlömer, O., Herget, J., Euler, T. (2020) Boundary condition control of fluvial obstacle mark formation – framework from a geoscientific perspective. Earth Surf. Process. Landforms, 45, 189–206.

Schwanghart et al., (2015) Repeated catastrophic valley infill following medieval earthquakes in the Nepal Himalaya. Science, 351, 147-150.

van Rijn, L.C. (2019) Critical movement of large rocks in currents and waves. International Journal of Sediment Research, 34, 387-398.

Whipple KX, Hancock GS, Anderson RS. (2000) River incision into bedrock: mechanics and relative efficacy of plucking, abrasion and cavitation. Geological Society of America Bulletin, 112, 490–503.
* * *

---

## Author Comment (AC1) · 2 Apr 2020

First of all, we thank Paul Carling for his interest in our study and also for taking the time to share his insightful suggestions and comments. This reply is aimed at providing key responses and details on the main points that were raised, in the spirit of Esurf discussions. These points will be addressed in more details in future iterations of this manuscript together with the points raised by other reviewers.

Perspective & paleo-hydraulics:

The first series of comments raised by P. Carling is related to the transport mechanisms

that mobilised these boulders. It is clear to us that at least some of these boulders have been transported over long distances. The lithology of the largest boulder in our dataset, for instance, can only be found outcropping 13 km upstream. This is the case for many of the surveyed boulders. It is true that the exact provenance of these boulders remains unclear, the fact that some of these boulders were previously incorporated in moraines before long-distance transport is indeed possible (as suggested in the comment), especially if these boulders are entrained by glacier lake outburst floods (GLOFs) as we suggest is a plausible explanation. However, there is a lack of evidence to directly support this hypothesis and, if correct, it would not change the conclusions of our study. Nonetheless, for the sake of completeness, we will mention this hypothesis in the discussion of our results in the revised version of the manuscript.

We do however not think that glacier transport to the location of deposition (< 1000 m a.s.l.), being it during the LGM or during earlier glaciations, is likely. There are very limited traces of the extent of the glaciations before the LGM on the southern flank of the central Himalayas (Owen & Dorch, 2014) but glacial chronologies from the lower relief, higher-elevation regions north of the range suggest that glaciers' extent during the Quaternary were not much larger than the LGM (Owen, 2020).

The hydraulics responsible for the transport of these boulders is necessarily poorly constrained as these types of flows are not frequently observed or instrumented. We, therefore, agree with P. Carling that the paleo-flow estimates we derive should be taken with caution but we sincerely believed this caution was conveyed by our manuscript; for instance, through the use of three available transport laws and our reporting of a broad 2 orders of magnitude range of what discharges might have been. We will emphasize these limitations further in the manuscript and will also refer to the references we might have missed that are cited in the short comment. It should nevertheless be noted that paleo-discharge estimate is not the finality of our work and that these estimates were made for comparison with available modern gauging data. This comparison suggests that paleo-discharges were most likely significantly larger than what is observed during

typical monsoons. And in some way, this confirms the obvious, since the fact that these boulders have been exposed for several kyrs in the river bed suggests that no recent flood has exceeded the flood that emplaced them.

We are not aware of creep processes that could explain slow boulder transport over long distances during monsoonal flows in such environments as is suggested in the comment. A travel distance of 10 km (shorter than generally observed from our data) over 5 kyrs would require about 2 m of movement per-monsoon season. This seems unlikely to us and would almost certainly result in greater dispersion of the exposure ages reported in our study.

As for the boulder size estimates, tape measurements have been made in the field when possible but these have proven challenging. Satellite imagery was therefore judged to represent a more homogeneous alternative. Overall boulder size estimates remain first-order estimates but these do not nuance our conclusion that emplacement has most likely occurred through exceptional hydraulic events.

Cosmogenic nuclides:

It is indeed true that our exposure age estimates represent minimum ages as is frequently the case with cosmogenic nuclides exposure dating and in the manuscript we acknowledge that erosion may lead to younger apparent ages. However, the general effect of surface erosion on boulder ages is relatively limited, especially for younger boulders. A surface erosion rate of 10 mm/kyr, as suggested, will change the calculated exposure age of ca. 150 years for our 5 kyrs boulders and about 1000 years for our older 12-15 kyrs boulder (Martin et al., 2017 - https://crep.otelo.univ-lorraine.fr). Surface erosion is therefore unlikely to significantly affect our exposure ages. As mentioned later, toppling of the boulders or the breaking of significant parts of the surfaces, if undetected during sampling, can affect boulder ages more significantly, but this would also result in stochastic ages.

It is possible that some of these boulders suffer from cosmogenic inheritance, i.e. pre

transport 10Be build-up linked to earlier exposure. This would result in boulder exposure ages significantly older than their true emplacement age. We do not observe such a pattern in our data as it is very unlikely that several boulders have received the same amount of pre-exposure, especially given the different lithologies implying different sources. Even though we do not favour the hypothesis that the boulder transport occurred slowly, over long time spans, such a mechanism would likely result in wide exposure age distributions, as boulders rotate during transport, successively exposing different sides for various amounts of time.

Even though the number of samples is limited in the Sunkoshi reach, we still identify three boulders out of five with ages compatible within uncertainty with the ages found by more boulders in the Trishuli, the remaining two being much older. It would have been beneficial to sample more boulders but we nevertheless believe that the suggestion of events occurring both in the Sunkoshi and the Trishuli around 5 kyrs is a robust finding of our study.

We hope that we have provided some constructive comments that help clarify the manuscript under discussion. As mentioned earlier we will incorporate these into an eventual future revision.

Marius Huber, Maarten Lupker & Sean Gallen on behalf of all co-authors

References

Martin, L.C.P., Blard, P.H., Balco, G., Lavé, J., Delunel, R., LIFTON, N., Laurent, V., 2017. The CREp program and the ICE-D production rate calibration database: A fully parameterizable and updated online tool to compute cosmic-ray exposure ages. Quaternary Geochronology 38, 25–49. doi:10.1016/j.quageo.2016.11.006

Owen, L.A., 2020. Quaternary Glaciation of the Himalaya and Adjacent Mountains, in: Dimri, A.P., Bookhagen, B., Stoffel, M., Yasunari, T. (Eds.), Himalayan Weather and Climate and Their Impact on the Environment, Himalayan Weather and Climate

and Their Impact on the Environment. Springer International Publishing, Cham, pp. 239–260.

Owen, L.A., Dortch, J.M., 2014. Nature and timing of Quaternary glaciation in the Himalayan-Tibetan orogen. Quaternary Science Reviews 88, 14–54. doi:10.1016/j.quascirev.2013.11.016
* * *

---

## Referee Comment (RC1) · Georgina Bennett (Referee) · 15 Apr 2020

This paper investigates the origin and timing of boulders found along two major rivers in Nepal. It uses cosmogenic exposure dating to date several boulders from each valley making up the largest grain sizes of river deposits in these valleys. It analyses their geology to identify potential source areas and thus travel distances of the boulders. Finally, it estimates the river discharges that would have been needed to transport the boulders using three different approaches. The authors find a clustering of ages around 5 Ky in both valleys. They also find that boulders have travelled over 10s of kms based on their geology. It calculates that discharges of 10ˆ3 – 10ˆ5 cubic meters

per second were needed to transport the boulders, exceeding the flows on record at discharge gages along these rivers but inline with estimates of several paleo lake outburst flood discharges. The authors discuss potential triggers of lake outburst floods, namely earthquakes (via generation of landslide dams) and climate (through generation of glacial lakes and outburst floods). The authors reason that boulders were likely transported by GLOFS that coincided with widespread glacial retreat around 5Ky in the region at a period of increased aridity. This suggests the significance of climate events in the evolution of these valleys.

I enjoyed reading this paper, which investigates the interesting and important question of the origin and transport history of large boulders, a prominent feature of mountainous valleys in the Himalaya. The clustering in boulder ages in the two valleys is interesting and hints at a large event that transported or exposed these boulders. If the trigger was indeed a series of glacial lake outburst floods related to widespread glacial retreat, this raises concerns about what may await such valleys under current warming conditions and glacial retreat. The paper is well written and presented and I think needs just a few minor/moderate revisions.

I made some comments on the manuscript (attached) and some of these may be duplicated here, but my main comments are as follows:

I wonder if you could validate your methods for backcalculating paleodischarges from mobilizing boulders of varying sizes with the event documented by Cook et al., 2018 in the Sunkosi? Cook et al. 2018 found that the event mobilized boulders as large as 5.7m. Additionally, discharge is known for this event. You could also investigate the distances moved of boulders in the 2016 event, if this information is available (I'm not sure if Cook et al, 2018 were able to track individual boulders in fact), and compare these with the distances you suggest your boulders moved in a single event.

I am wondering whether these boulders could have been moved by successive events and that potentially a particularly large event around 5000 years exposed these? For

example, could these have been transported in an older event or a series of older GLOFs, buried, e.g. in a terrace and then exposed around 5000 years ago for example by erosion of the surrounding terrace? Perhaps more potential storylines should be discussed in light of uncertainties.

Whilst you emphasize the clustering in time of your boulder dates, there seems to be a lot of spread in ages of other dated boulders in the region, particularly in the Everest region (Figure 6). More discussion of this spread would be good as it perhaps goes against the idea of a synchronous GLOF response to glacier retreat.

I also discussed this paper in our regular group reading session and the following additional comments came up:

Could you include potential source zones in Figure 1 based on the geology of the region? Could there be local, undocumented intrusions that these boulders could originate from? (this came from a group member who had actually visited the boulder in Figure 2D with Nepalese scientists)

Would it be possible to reconstruct the size of lake needed to generate a flood of the range of magnitudes you predict with your paleo flood estimates? Is there evidence in the topography that such a lake could have existed i.e. in terms of available accommodation space? Similarly, is there evidence that glacial lakes today are not completely filling this available space and hence may generate smaller magnitude GLOFS than in the past, as you suggest in the discussion?

Finally, I have just read the Paul Carling's comments and the authors' response before uploading my review (I'm not sure if this is common practice or not, being a relatively new ESurf reviewer!) I will just make the comment I agree with the authors that a creep mechanism of boulders is highly unlikely for boulders >2m, based on observations of boulders in the Upper Bhote Koshi by Cook et al (2018) who observed no movement of boulders during monsoon floods; only in the 2016 GLOF did boulders >2m actually move.

Please also note the supplement to this comment:
https://www.earth-surf-dynam-discuss.net/esurf-2020-17/esurf-2020-17-RC1-
supplement.pdf

---

## Referee Comment (RC2) · Anonymous Referee #2 · 17 Apr 2020

Review of Huber et al. "Timing of exotic, far-travelled boulder emplacement and paleo-outburst flooding in the central Himalaya" – ESurf, April 2020.

Huber et al. examine a series of large boulders in the central Himalaya. The lithologies of these boulders suggest that they are sourced from further upstream, over 40 km in some instances, implying they have undergone significant transport to reach their current location. Using cosmogenic radionuclide exposure dating of the boulders, Huber et al. find their ages tend to cluster, with the most notable clustering in ages being around the Early Holocene Climatic Optimum, ~5 ka. They infer that these boulders have been transported by large glacial lake outburst floods (GLOFs), which may have

been more prevalent in the region at this time as climate transitioned into more arid conditions that would have promoted widespread glacier retreat. Using digitized topographic maps to determine valley-cross sections and gradients, peak discharge values that would have been required to mobilise these boulders were calculated using 3 different methods. The authors conclude that the magnitude of flow to transport these boulders is larger than what could be achieved by modern monsoonal discharges, but comparable to other documented GLOFs. The authors comment on the wider importance of this work in terms of the interactions between large, infrequently mobilized boulders and bedrock incision patterns and longer-term landscape evolution, as well as understanding climatic controls on lake outburst flood frequency and magnitude.

In general, the manuscript is well written and presents a very interesting story – it is not uncommon to find these large boulders dotted around the Himalaya so it is nice to finally see an analysis of how and where they have come from. It also serves as an important contribution to understanding how rare but large magnitude discharge events contribute to landscape evolution, and how this may link to broader patterns of climate change. I do feel that the manuscript would benefit from a few additional clarification and considerations, and I hope the comments below help generate some discussion.

General comments:

One aspect that was/is not obviously clear to me is where these boulders have originated from. Are these deposits which would have been sat in the valley of their source lithology (e.g. delivered from local hillslope wasting) and then transported by large flows? Or are these boulders that would have been glacially eroded and then exposed during glacial retreat and subsequently flushed out? I would have thought the latter would be less likely as subglacially sourced rock would be highly fractured. If there had been some CRN accumulation on the valley floor prior to entrainment, this may help explain some of the spread in boulder ages. Looking at Figure 4, there is still a reasonable spread around a series of boulders ($\sim$2.5-6 ka) which are suggested by the authors to have been transported by a single event. While some of that spread can

be attributed to error, some could also be explained by pre-transport accumulation, or a whole range of other factors (e.g. intermittent transport by a series of events rather than a single one). I felt this discussion was lacking from the main text.

Figure 1 would help tie a lot of this information together better if it had 1) the different lithologies mapped out, 2) the corresponding boulder lithologies shown, 3) best-guess of where paleo-ice limits would have been. Perhaps even make this as a separate figure zoomed in.

I'm curious as to why we don't see any boulders relating to older glaciations/climatic changes, if this is the proposed mechanism of transport. Have these boulders been buried? Was this most recent event so large that it flushed everything else through – and are there more boulders further downstream? One aspect I'm struggling to visualize is the local valley conditions in which these boulders are deposited – are they typically found in regions of local valley widening for example, where you may feasibly be able to temporarily store/bury larger clasts. Some details may be helpful.

One additional comment here would be whether boulder emplacement could be by debris-flows, which in theory could raft very large particles a considerable distance. A number of moraine dam failures in Clague and Evans (2000) resulted in debris flow initiation, transporting boulders several kilometers, despite relatively modest initial discharges from the lakes. If the boulders represent the coarsest material within a much older deposit that has reset the CRN clock, finer material may have been washed away leaving a layer of surficial boulders (almost like a deflation layer I guess?) with potentially quite similar ages. Presumably these valleys have gone through considerable phases of aggradation and incision? This reminds me of one particularly large boulder further downstream on the Koshi just south of Chatara – lat 26.813896, lon 87.152834. The boulder is maybe 5-6 m in diameter and sat in the middle of a sand-bed river – it must have been transported several km (over a very low gradient) to get there, yet there are no other deposits nearby. Is it just the remnants of an ancient mass flow deposit that spilled out of the mountain front and everything else has been washed away?

Given the large transport distances of some of these boulders – would some abrasion/fracturing have occurred during transport? Presumably boulders would have been moving/saltating on the bed which is likely to produce quite high rates of abrasion/mass loss, so initial boulder sizes may have been a good chunk larger. Some comment to that effect may be helpful. If these boulders are of glacial landscape origin, this may also explain the absence of glacial features/signatures on their surface.

Flow estimates – To transport boulders of the sizes you have, how long would these flows need to persist in order to transport the minimum distances? I'm just looking at Figure 3a – you'd need a peak discharge sustained for nearly an hour or two to transport some of those boulders the required distances. Is that reasonable (I have no idea!)? Or is it more likely that these boulders would have been moved progressively (and potentially rotated)? An estimate or rough calculation of paleo-lake size to generate a discharge of that magnitude may be useful here. I'm not hugely familiar with the characteristics of proglacial lakes, but how big do moraine-dammed lakes tend to be in comparison to ice or landslide dammed ones? The magnitude of discharges you predict here are large – could you feasibly hold that much water behind a moraine-dammed proglacial lake?

Specific comments:

P1L11: First line of abstract 'commonly linger in Himalayan river channels'

P1L23: Why is this counter-intuitive? If climate is warming (as you state) then you'd expect more melt? You later mention that past large phases of glacier retreat have been suggested to increase GLOF frequency, so it doesn't seem especially counter-intuitive! It seems the only mention of increased temperature is in the abstract – so which is happening/more important – increased temperature (+ more melt) or aridity? Presumably, a case simply of increased aridity doesn't necessarily mean increased melt unless accompanied by some change in temperature. Some further explanation in the text further down may help. How exactly – is it simply that there are more moraines

exposed so more damming of the usual meltwater?

P1L31: Earthquakes are pretty common in the Himalaya – specify the sizes/magnitudes that are of importance.

P2L39: Worth noting that not all proglacial/landslide dammed lakes drain catastrophically so there are probably more of these lakes than there are catastrophic events.

P2L46: How far back do these records go?

P3L76: What about even older glaciations? Just a quick comment to say how they fit in.

P4L127: Why was boulder diameter measurement from Google Earth as opposed to field measurements? What is the uncertainty/resolution/error? I'm confused about how you get an intermediate axis from a satellite image of a boulder where some of the boulder may be buried – it's hard enough on a high resolution photo. More details are needed.

P5L131: What is the vertical spacing of the contours on these maps...50 m? Does this give better valley geometry than any of the available DEMs? Is there information relating to the modern channel from any of the DHM stations which could be used to help improve (at least a part of) these cross-sections?

P5L135: Were any of these calculations done for valley geometry further upstream near the source lithologies, where the boulders were actually entrained? Presumably if everything has been calculated using digitized maps this could have been looked at – so for the same discharge in the upstream valley geometry, what size boulder could have been transported/motion initiated, does it match? I'm also curious about where the biggest source of uncertainty comes from in the paleo-discharge calculations. You comment on this in section 5.2, stating that the channel geometry is probably one of the biggest uncertainties. Are the results more sensitive to uncertainties in slope or cross-sectional area, for example? Even if paleo valley estimates were available, there

is still no constraint on how the bed or valley floor may have evolved during the event.

P7L213: Could these abrasion marks also have been generated while the boulder has sat in-situ during subsequent high flows where the boulder has been submerged/partially submerged? If this is a possibility, have any of the exposure ages been corrected for post-depositional erosion? Some comments on this and potential effects on exposure ages would help.

P11L345: Yes, but not all earthquakes will generate LLOFs – on the flip side, presumably a modest earthquake at the right time in the right place could also generate a LLOF?

Figure 2: I find the red text quite hard to read – could you put a white box behind?

Figure 6: Do you see boulders within ∼50 km of the moraine deposits in any of these other catchments?

[Figure]

---

## Author Comment (AC2) · 22 Apr 2020

We thank Georgina Bennett for her interest in the study and the constructive comments she provided. We overall note that her comments (as well as from the other reviewer) require an improved and broader discussion of alternative transport processes and emplacement mechanisms for the studied boulders. We will include these remarks, along with all other, in a revised version of the manuscript once the discussion period closes and depending on the associate editor's suggestions.

Marius Huber, Maarten Lupker & Sean Gallen, on behalf of all co-authors.

[Figure]

**ESurfD**

---

## Author Comment (AC3) · 22 Apr 2020

We would also like to thank the second reviewer for her/his comments. Similarly to the first review, these comments were well received, constructive, and triggered some discussion amongst us. We will incorporate the results of these later in a revised version of the manuscript. This new version will hopefully clarify the discussion about the range of possible transport mechanisms of these boulders but will probably also highlight some of the limitations of our data in answering all the questions that were raised.

Marius Huber, Maarten Lupker & Sean Gallen, on behalf of all co-authors.

[Figure]

**ESurfD**

Interactive
comment

---

## Author Response (AR1)

Marius Huber

Centre de Recherches Pétrographiques et Géochimiques

15 rue Notre Dame des Pauvres BP 20

54500 Vandœuvre les Nancy, France

Dr. Wolfgang Schwanghart

Institut für Umweltwissenschaften und Geographie

Karl-Liebknecht-Str. 24-25

14476 Potsdam-Golm, Germany

**Revised submission**

| | |
|---|---|
| Scientific journal: | Earth Surface Dynamics (ESurf), EGU Publications |
| Contribution title: | Timing of exotic, far-travelled boulder emplacement and paleo-outburst flooding in the central Himalaya |
| Authors: | Marius L. Huber, Maarten Lupker, Sean F. Gallen, Marcus Christl, and Ananta P. Gajurel |
| Received: | 28 Feb 2020 |

Dear Dr Schwanghart,

We got valuable responses in the form of one 'Short comment' and two Referee replies, which lead us to revise parts of our manuscript:

- **Short Comment from Paul A. Carling: (SC 2)**
- **Referee comment by Georgina Bennett (RC1), Referee #1**
- **Referee comment by Anonymous Referee (RC2), Referee #2**

The received comments and suggestions have been constructive and we feel that they have greatly helped clarify our study. The main concern that was shared by all the reviewers, about the need for a more in-depth and nuanced discussion of the possible boulder emplacement mechanisms was addressed by rewriting a section of the discussion. Most other comments were also implemented and are detailed in the following parts of the response letter. More minor and secondary comments are not further detailed in this letter but have almost always been included in the revised manuscript. All modifications are documented in an attached marked-up document the changes that were made.

We would like to thank all the reviewers and the e-surf editorial board for the efficient handling and constructive comments received so far.

Marius Huber, on behalf of all the co-authors.

Reviewer's comment

*Answer to comment*

Modification in the manuscript

**Referee #1 (G. Bennett)**

[1] I wonder if you could validate your methods for backcalculating paleodischarges from mobilizing boulders of varying sizes with the event documented by Cook et al., 2018 in the Sunkosi? Cook et al. 2018 found that the event mobilized boulders as large as 5.7m. Additionally, discharge is known for this event. You could also investigate the distances moved of boulders in the 2016 event, if this information is available (I'm not sure if Cook et al, 2018 were able to track individual boulders in fact), and compare these with the distances you suggest your boulders moved in a single event.

*This is a great idea! We have included this comparison with the 1985 GLOF water velocity and mobilised boulder sizes data from Xu, 1988 and with the 2016 GLOF data reported in Cook et al. (2018). Both comparisons show a good agreement with the transport laws that were used in the manuscript.*

**Section 5.2, p.9:** These estimates are corroborated by observed boulder movement under known discharges (Figure 3A) reported in Xu (1988) and Cook et al. (2018) in the upper Sunkoshi (some 30 km upstream of Balephi). During the 2016 GLOF event, Cook et al. (2018) report the movement of a ca. 5.7 m diameter boulder for mean flow velocities between 8.2 and 6.8 m/s. An earlier study also reported the movement of a boulder of ca. 11.3 m in intermediate diameter for water flow velocities between 8.4 and 8.0 m/s during the 1981 GLOF in the same reach (Xu, 1988). Velocity and discharge estimates broadly agree with our estimates derived from the Costa's (1983), Clarke's (1996) and Alexander and Cooker's (2016) relations for boulder incipient motion. It is important to mention that peak discharges cannot be directly compared because of the various distance to potential source areas.

**Updated Figure 3:**

[Figure]

**Figure 3: A- Theoretical flow velocities required to move boulders of a given diameter with explanatory input parameters (channel bed slope 0.03, rock material density 2700 kg/m3, fluid density 1500 kg/m3) according to the parametrisations and models of Costa (1983), Clarke (1996) and Alexander and Cooker (2016). The grey shaded area indicates range of boulder intermediate diameters from this study. Green and red rectangles are bounded by velocity estimates upstream and downstream boulders (5.7 m and 11.3 m in diameter) mobilised during the 2016 and 1981 GLOF events in the upper Sunkoshi (Cook et al., 2018; Xu, 1988). The Clarke (1996) method is plotted with a channel bed slope adjusted to Cook et al. (2018) which is the gradient of the Sunkoshi reach at the location of boulder movement (0.0245). B- Estimated peak-paleo-discharges required to move the studied boulders according to the three models that were used for paleo-discharge calculations. Green and red rectangles are bounded by upstream and downstream estimates of observed boulder movements for the 2016 and 1981 GLOF events in the upper Sunkoshi (Cook et al., 2018; Xu, 1988).**

 Whilst you emphasize the clustering in time of your boulder dates, there seems to be a lot of spread in ages of other dated boulders in the region, particularly in the Everest region (Figure 6). More discussion of this spread would be good as it perhaps goes against the idea of a synchronous GLOF response to glacier retreat.

*We have updated the discussion on the observed spread in exposure ages. These are likely mainly linked to a) the nature of the very dynamic fluvial system that is unlikely to preserve surfaces very well and may induce some erosion, b) exposure or shielding before final emplacement and c) for some boulders the spread is linked to their geomorphic position in the system (younger boulders in the stream, vs older boulders on the small fill terrace, which we interpreted as the result of the fill terrace re-incision). It is not clear to us however, what comparison can be made with boulders on figure 6 from the Everest region, as these are glacially transported moraine boulders and are hence are more stable geomorphic features compared to boulders exposed in the river channel.*

**Section 5.3, p.11:** However, as we noted earlier, the upstream Trishuli reach, boulders in the active channel are systematically younger (1-2 kyrs) compared to boulders located on the adjacent terrace. We interpret these younger ages as the result of the shielding of the boulders trapped in this thin fill deposit (< 20m thickness) before being exposed when the river re-incised the deposit (Figures 2 and 4; Table 1; S1). This re-incision had, however, to occur rapidly, or the age difference between in-channel and terrace-top would be larger.

**Section 5.3, p.12:** The relatively large spread of exposure ages compared to other settings such as moraine boulders, for instance, can likely be attributed to the fluvial setting. Boulders can be affected by nuclide inheritance, if surfaces were exposed for significant durations prior to entrainment, which would bias the ages too old. Erosion of the boulder surface, a plausible process for boulders sitting in the channel of a mountainous stream, would bias the ages too young. But only significant erosion, such as the fracturing of a sizeable part of the boulder surface (which was avoided for sampling if recognized in the field) would affect the exposure ages (a steady-state weathering of the boulder surface of 10 mm/kyr would reduce the exposure age of a 5 kyrs old boulder by ca. 150 yr). Partial cover by sediments is also a plausible explanation for the scatter in exposure ages, and we invoke this effect for the younger ages of the boulders in the channel of the upstream reach of the Trishuli or boulder NEQ/162 59. However, since these processes (inheritance, erosion or partial sediment cover) are stochastic, it would be expected that if they dominated the signal, exposure age distributions would be more widespread then what was observed.

 Could you include potential source zones in Figure 1 based on the geology of the region? Could there be local, undocumented intrusions that these boulders could originate from? (this came from a group member who had actually visited the boulder in Figure 2D with Nepalese scientists).

*A proper rendering of the possible source regions on the Figure 1 is complicated by the unprecise geology and mixture of lithologies. All gneissic lithologies can be found structurally above the Main Central Thrust (MCT) based on all published work in the area (and elsewhere in the Himalaya). The MCT is clearly marked and referred to in the text and the figure. We feel that a more precise source region is too uncertain to report here. Gneissic lithologies in the immediate vicinity of the sampling locations need to be present in the form of local klippe of High Himalayan Crystalline units in what is otherwise the Lesser Himalaya. These have not been reported or observed but cannot be formerly ruled out. We however argue that a local emplacement is unlikely as a number of boulders of different lithologies have similar exposure ages and are therefore unlikely to be all locally derived by a major landslide for instance (we would expect a similar lithology in such a case). This has been clarified in the main text:*

**Section 5.1, p.8:** Diagnostic mineralogy and fabric of the other surveyed ortho-gneiss boulders in the Trishuli catchment are not present in the Lesser Himalayan sequence and, therefore, must originate from areas upstream (or structurally above) of the MCT (Figure 1). No known Higher Himalayan unit klippe is mapped on hillslopes directly above the studied reaches, and the mixed boulder lithologies make a local emplacement source through mass-wasting unlikely. These observations, therefore, require minimum transport distances of approximately 22 km to 46 km depending on the present boulder location (Table 1).

[4] Would it be possible to reconstruct the size of lake needed to generate a flood of the range of magnitudes you predict with your paleo flood estimates? Is there evidence in the topography that such a lake could have existed i.e. in terms of available accommo- dation space? Similarly, is there evidence that glacial lakes today are not completely filling this available space and hence may generate smaller magnitude GLOFS than in the past, as you suggest in the discussion?

*We would have liked to follow this path but quickly felt that this was out of the scope of the present work. The recent literature suggests that modelling of these flood flows in rough topography such as Himalayan valleys requires advanced modelling approaches to yield realistic estimates (e.g. Turzewski et al., 2019 – JG Earth Surface). Furthermore, these modelling approaches would have to be systematically applied to explore the flood propagation from a large number of potential locations in the upstream valleys and include the variability in breach mechanisms. This would require dedicated additional work that does not fit in the present paper. It is however definitely something worth exploring in the future.*

**Section 5.2, p.10:** The reconstruction of flood duration or initial lake size is, however, hampered by the multiple possible lake locations, unknown breach mechanisms, and the large uncertainties of our paleo-flood discharge estimates. Such a reconstruction is beyond the scope of this work and would require dedicated and computationally expensive fluid flow numerical models (e.g. Carling et al., 2010; Denlinger and O'Connell, 2010; Turzewski et al., 2019).

*Further specific comments were all addressed and adequate changes made to the manuscript.*

**Referee #2 (Anonymous)**

[5] One aspect that was/is not obviously clear to me is where these boulders have originated from. Are these deposits which would have been sat in the valley of their source lithology (e.g. delivered from local hillslope wasting) and then transported by large flows? Or are these boulders that would have been glacially eroded and then exposed during glacial retreat and subsequently flushed out? I would have thought the latter would be less likely as subglacially sourced rock would be highly fractured. If there had been some CRN accumulation on the valley floor prior to entrainment, this may help explain some of the spread in boulder ages. Looking at Figure 4, there is still a reasonable spread around a series of boulders (~2.5-6 ka) which are suggested by the authors to have been transported by a single event. While some of that spread can be attributed to error, some could also be explained by pre-transport accumulation, or a whole range of other factors (e.g. intermittent transport by a series of events rather than a single one). I felt this discussion was lacking from the main text.

*We would agree with the reviewer that many questions about the provenance and emplacement mechanisms of these boulders remain enigmatic. We fear however that we cannot fully answer all these questions about the exact configuration of these boulders before transport based on the available data without unfounded speculations. The entire discussion reviewing the possible transport and emplacement scenarios has been re-written so as to highlight a number of possible (but in our opinion unlikely) alternative hypothesis:*

[revised manuscript text omitted]

*The discussion about the possible explanations for the spread in the data has also been refocused (as already mentioned in the reply to comment #2 from G. Bennett.*

[6] Figure 1 would help tie a lot of this information together better if it had 1) the different lithologies mapped out, 2) the corresponding boulder lithologies shown, 3) best-guess of where paleo-ice limits would have been. Perhaps even make this as a separate figure zoomed in.

*As already replied to comment #3, we have decided against adding a detailed geological map to figure 1. This would both be imprecise and difficult to read. The main lithological information required to justify the far travelled nature of the studied boulders is reported in the form of the trace of the Main Central Himalayan Thrust (MCT) which separates the mostly gneissic units of the High Himalayan Crystalline (HHC) rocks to the North from the mostly schist and quartzite to the South. Therefore, all gneissic boulders that were surveyed have travelled at least the distance separating them from the MCT. Any further attempt to link the boulder lithology to specific regions in the HHC would require a detailed mapping beyond what is available or has been carried out.*

*The traces of past glacial extents on the southern slopes of the Himalayan range are elusive and poorly preserved. Holocene moraines that were used in figure 6 to argument for a phase of glacial retreat at ca. 5 kyrs, are found within a few km of present glacier extent and would not be very visible on a map of this scale. Older LGM and Quaternary glaciations did likely not result in much larger glacier extents compared to present based on the evidence from the northern flank of the Himalayan range, but their exact extent to the South of the range is unclear. We have nevertheless added to Figure 1 the proposed maximum Quaternary glacier extent proposed by Shiraiwa and Watanabe (1991) for the Langtang valley (upstream of the Trishuli boulders) based on landscape morphology and valley shape. It should be noted that, at 2500 m of elevation, this Quaternary glacial limit is still ca. 2 km above the boulder location. We believe that these arguments should completely exclude the possible glacial transport of these boulders to their present location during the LGM or earlier.*

**Section 5.1, p. 8:** As noted above, boulders in both valleys are well below the extent of alpine glaciers in the modern or during the last and previous glacial maximum stages and their associated glacial deposits (e.g. Shiraiwa and Watanabe, 1991; Owen and Benn, 2005; Owen and Dorch, 2014; Owen, 2020; Figures 1 and 2). The low elevations where the exotic boulders are presently observed excludes a glacial transport mechanism. Rather, the observed locations and our provenance analysis indicate that the mobilization and transport of large grain-sizes occurred in central Himalayan river valleys over long distances (>10 km), most likely through fluvial processes.

[7] I'm curious as to why we don't see any boulders relating to older glaciations/climatic changes, if this is the proposed mechanism of transport. Have these boulders been buried? Was this most recent event so large that it flushed everything else through – and are there more boulders further downstream? One aspect I'm struggling to visualize is the local valley conditions in which these boulders are deposited – are they typically found in regions of local valley widening for example, where you may feasibly be able to temporarily store/bury larger clasts. Some details may be helpful.

*This is a good point and we think that it is likely related to the limited survival of boulders in mountain river streams. Large boulders are likely stable in mountain streams for quite a while, only slowly being abraded by fluvial processes until they reach a threshold size bellow which they become easily mobilised and essentially disappear from the fluvial bed (by more rapid abrasion during transport). We are not aware of any specific work that quantifies these boulder survival durations even though it is an interesting (and likely important) question. We have referred to this in the text and reference modelling work that indirectly addresses these concepts but can only speculate about the absence for older boulders.*

**Section 5.3, p. 11:** It should also be noted that there is a likely upper limit to the survival duration of boulders in a fluvial channel, as fluvial abrasion and comminution processes during transport (Attal and Lavé, 2009; Carling and Fan, 2020) or while at rest in the channel bed (Shobe et al., 2016; Glade et al., 2019) will ultimately reduce the size of the boulder until it can be exported by more frequent smaller flows.

[8] One additional comment here would be whether boulder emplacement could be by debris-flows, which in theory could raft very large particles a considerable distance. A number of moraine dam failures in Clague and Evans (2000) resulted in debris flow initiation, transporting boulders several kilometers, despite relatively modest initial dis- charges from the lakes. If the boulders represent the coarsest material within a much older deposit that has reset the CRN clock, finer material may have been washed away leaving a layer of surficial boulders (almost like a deflation layer I guess?) with po- tentially quite similar ages. Presumably these valleys have gone through considerable phases of aggradation and incision? This reminds me of one particularly large boulder further downstream on the Koshi just south of Chatara – lat 26.813896, lon 87.152834. The boulder is maybe 5-6 m in diameter and sat in the middle of a sand-bed river – it must have been transported several km (over a very low gradient) to get there, yet there are no other deposits nearby. Is it just the remnants of an ancient mass flow deposit that spilled out of the mountain front and everything else has been washed away?

*The transition from a "clear water" flow to a debris flow is likely continuous and the sediment concentration cannot be reconstructed. As we note in the main text (section 4.2, p.7) our calculations have been made with a flow density of 1.5 to account for a high sediment load. We also noted that fairly high sediment concentration flows can still be reasonably approximated by a Newtonian turbulent flow law, but added that the flow discharges would be inaccurate for pure debris flows.*

**Section 5.2, p. 8:** In hyper-concentrated flows with 40 to 70 wt. % sediment entrained, non-Newtonian, plastic fluid behaviour and laminar flow can arise due to the establishment of shear strength in the fluid material (e.g. Pierson and Costa 1987). However, if the amount of sediment entrainment remains at the lower end of this "hyper-concentrated" range, flow mechanics are still adequately approximated by Newtonian, turbulent flow of a "clear" waterflood (Costa, 1984; Pierson and Costa, 1987; Pierson, 2005; Wang et al., 2009; Hungr et al., 2014) as was assumed here. Nevertheless, our calculations do not apply for higher sediment load conditions, for example, conditions associated with debris flows.

*Our flow estimates are therefore bound to this flow assumption and we included a necessary cautionary note while discussing them. The comparison of the transport laws we used with literature flow and boulder mobilisation data (as suggested by Referee #1) suggests that our assumptions are reasonable. But maybe more importantly, we want to emphasize that the possible debris flow nature of the flows that emplaced the boulders, does not change our conclusion that these flows are exceptional and rare which is the main discussion point of the paper (rather than the actual paleo-discharge estimate itself).*

*The boulder pointed at by the reviewer in the floodplain of the Koshi is indeed visible on google earth. If the reviewer sampled it and would be ready to share his sample, we would be happy to provide him with an exposure age.*

[9] Given the large transport distances of some of these boulders – would some abra- sion/fracturing have occurred during transport? Presumably boulders would have been moving/saltating on the bed which is likely to produce quite high rates of abrasion/mass loss, so initial boulder sizes may have been a good chunk larger. Some comment to that effect may be helpful. If these boulders are of glacial landscape origin, this may also explain the absence of glacial features/signatures on their surface.

*This point is remains speculative in our opinion, but it is likely that boulders are continuously abraded, both during the transport as well as at rest in the river. We quote the same paragraph as for comment #7.*

**Section 5.3, p. 11:** It should also be noted that there is a likely upper limit to the survival duration of boulders in a fluvial channel, as fluvial abrasion and comminution processes during transport (Attal and Lavé, 2009; Carling and Fan, 2020) or while at rest in the channel bed (Shobe et al., 2016; Glade et al., 2019) will ultimately reduce the size of the boulder until it can be exported by more frequent smaller flows.

[10] Flow estimates – To transport boulders of the sizes you have, how long would these flows need to persist in order to transport the minimum distances? I'm just looking at Figure 3a – you'd need a peak discharge sustained for nearly an hour or two to transport some of those boulders the required distances. Is that reasonable (I have no idea!)? Or is it more likely that these boulders would have been moved progres- sively (and potentially rotated)? An estimate or rough calculation of paleo-lake size to generate a discharge of that magnitude may be

useful here. I'm not hugely famil- iar with the characteristics of proglacial lakes, but how big do moraine-dammed lakes tend to be in comparison to ice or landslide dammed ones? The magnitude of dis- charges you predict here are large – could you feasibly hold that much water behind a moraine-dammed proglacial lake?

*This point is similar to the point #4, from Referee 1. For the same reasons we have not made such an attempt as lake volume, flow duration etc.. would be difficult to reconstruct without a dedicated flow model, a detailed study of possible lake locations and assumptions on lake breaching mechanisms. We feel that these are beyond the scope of the current paper but would make for a very interesting follow-up study.*

*Further specific comments were all addressed and adequate changes made to the manuscript.*
* * *
**Interactive comment by Paul Carling (25.03.2020).**

*We have provided an answer to the most remarks and suggestions that were raised by Paul Carling in his online comment directly in an earlier online response. These will not be repeated here, but we highlight some of the changes that were made to the manuscript as a consequence.*

[11] Palaeohydraulics using boulder data

*The discussion on the entrainment laws that were used was expanded in the main text but we have not provided a full review of all available entrainment laws available, as it is not the main objective of this work. The addition of the validation of the flow laws estimates with literature data as suggested by Reviewer #1 is in our opinion a good indication that the formulations that were used provide reasonable estimates (see response to comment #1). Also as highlighted in the earlier response to P. Carling, we consider that the paleo-discharge estimate is not the main finding of this paper, but rather strengthen the conclusion that events emplacing these boulders are rare and of high magnitude.*

[12] Cosmogenic dating

*P. Carling raised comments about the spread in the exposure ages and the potential for other emplacement scenarios. We did not change our views on the most likely emplacement scenario through single, large magnitude events, but provide a re-written discussion of the possible alternative scenarios and causes for the spread in exposure ages (see reply to comment #2 and comment #5).*

[revised manuscript text omitted]

**Commented [A8]:** Modification in response to Short comment by Paul A. Carling and "specific comments" by Referee #2. References added:

[revised manuscript text omitted]

180
* * *
**Commented [A11]:** Referee #2 requested more details about the boulder diameter determination. Paul A. Carling mentioned some drawbacks associated with our method of boulder determination using Google Earth imagery in his Short comment, too.

**Commented [A12]:** RC2 "Specific comments": see modified supplement for more details

**Commented [A13]:** Addition in response to Referee #2's "specific comments" and a remark by Referee #1 (Bennett).

[revised manuscript text omitted]

Additionally, Paul A. Carling mentioned in his Short comment the possibility of boulder entrainment from glacial deposits and a possible significant reduction in fluvial travel distances considering glacial transport processes. By referring to the given references and Figure 1 and 2, a significant decrease in assumed fluvial travel distances can be excluded.

**Commented [A17]:** After remarks in Paul A. Carlings Short comment and from Referee #2 regarding debris flow transport we want to clarify explicitly that our method for peak-discharge estimation does not apply for flow conditions usually referred to as "debris flow conditions" in literature.

crudely represented and do not account for past channel morphologies before and during the time of the floods. Since detailed riverbed morphology is required for the hydraulic discharge calculation, additional uncertainty arises from the resolution of the data used here and the necessity of using the modern channel geometry for these calculations. We hypothesize that these uncertainties are the main reason for the discrepancies between paleo-peak-discharge estimates for boulders from a similar age range that were presumably moved during a single event (Figure 3). First-order discharge estimates for boulder transport of surveyed clast sizes therefore broadly range from ca. $10^3$ to $10^5$ m³/s.

These estimates are corroborated by observed boulder movement under known discharges (Figure 3A) reported in Xu (1988) and Cook et al. (2018) in the upper Sunkoshi (some 30 km upstream of Balephi). During the 2016 GLOF event, Cook et al. (2018) report the movement of a ca. 5.7 m diameter boulder for mean flow velocities between 8.2 and 6.8 m/s. An earlier study also reported the movement of a boulder of ca. 11.3 m in intermediate diameter for water flow velocities between 8.4 and 8.0 m/s during the 1981 GLOF in the same reach (Xu, 1988). Velocity and discharge estimates broadly agree with our estimates derived from the Costa's (1983), Clarke's (1996) and Alexander and Cooker's (2016) relations for boulder incipient motion. It is important to mention that peak discharges cannot be directly compared because of the various distance to potential source areas.

To place our results in the context of previous studies, we compare our discharge to those from the literature and historical records. Cenderelli and Wohl (2001) compared seasonal high flow floods (SHFFs) with discharges of recent GLOF events and they appeared to be at least one order of magnitude higher than monsoonal precipitation peak discharges in the central Himalayan Mount Everest region for reaches spanning many 10's of kilometres downstream of the breach locations. Peak discharges reaching $10^5$ m³/s substantial distances downstream have been documented or suggested for a few historical events in the Himalaya mainly associated to LLOF such as the Great Indus flood of 1841 (Mason, 1929; Shroder et al., 1991), the great outburst in April 2000 in the Tibetan Yigong Zangbo River (Shang et al., 2003; Delaney and Evans, 2015; Turzewski et al., 2019), and the large LLOFs at Dadu River and Yalong River in the years 1786 and 1967 in Sichuan province, China (Dai et al., 2005; Runqiu, 2009). These events are, however, rarely observed even though there is sedimentological evidence that large-scale LLOF events happened regularly throughout the Holocene within the same catchments (e.g. Hewitt et al., 2011; Wasson et al. 2013). To our knowledge, GLOF discharge estimates of historically documented events in the Himalaya reach ca. $10^4$ m³/s (e.g. Vuichard and Zimmermann, 1987; Hewitt, 1982; Xu, 1988; Yamada and Sharma, 1993; ICIMOD et al., 2011; Cook et al., 2018) with Holocene reconstructed discharge estimates that exceeded $10^5$ m³/s, such as that reconstructed by Montgomery et al. (2004) for the Tsangpo River gorge outburst flood for locations >10 km downstream of the paleolake.

Hydrological stations from the Department of Hydrology and Meteorology, Government of Nepal allow comparison of our paleo-discharge estimates to measured discharges over the last decades. For the Trishuli reach, station number 447 (N27.97, E85.18) near the town of Betrawati, is located in between the two studied upstream and downstream boulder fields (Figure 1).

**Commented [A18]:** Addition following discussion in Referee #2's "Specific comments"

**Commented [A19]:** This paragraph was written in response to Referee #1's (Bennett) suggestion to validate our paleo-hydrology estimates with results obtained by Cook et al. (2018). We additionally compared our values with water flow estimates by Xu (1988) from an earlier GLOF that happened in the year 1981 also in the upper Sunkoshi. See new addition to the supplement (Figure S2).

**Commented [A21]:** Modification by the authors following reasoning by Paul A. Carling mentioned in his Short comment. This is an important clarification for our discharge estimates. Estimates should always be specified according to where they were made downstream from the breach. Peak discharges at breach locations are always much higher and decrease downstream due to attenuation of the flood wave.
Same below.

[revised manuscript text omitted]

**Commented [A24]:** This piece of discussion was added in response to questions raised by Referee #1 (Bennett) and Referee #2 which relate to flood duration time and outburst lake volume. Unfortunately, they cannot be dealt with in the scope of this contribution.

[revised manuscript text omitted]

**Commented [A30]:** This section was rewritten in response to Paul A. Carlings Short comment and both Referee comments, Referee #1 (Bennett) and Referee #2.
A more elaborate discussion expounding on boulder transport and age clustering was demanded by the readers. Now we provide more insight to our interpretations, referring to specific sampling sites in greater detail.

[revised manuscript text omitted]

**Commented [A38]:** Modified Figure 1:

Referee #1 (Bennett) and Referee #2 suggested changes to this figure: Referee #1 and #2 suggested adding potential source zones of the surveyed boulders based on geological mapping to the figure.
In the authors opinion Figure 1 provides already information on potential source zones by the MCT we included previous manuscript. The MCT separates Higher Himalayan rocks from Lesser Himalayan lithologies. Including geological mapping would result in a less readable figure which the authors tried to avoid. We decided against the Referees advice and did not add geological mapping to Figure 1.

Further Referee #2 proposed to add boulder lithologies. Figure 1 could be expanded by zoom-in figures or zoom-in figures could be added to the supplement.
The authors would like to draw attention to the lithology information given in Table 1, coordinates and the supplement material already provided (including Google Earth imagery of all the sampling locations and additional figures for the Trishuli catchment).
We decided not to expend Figure 1 or our supplement materials with additional zoom-in figures. The benefit of zoom-in figures is limited in our opinion and the supplementary materials are already voluminous.

Paul A. Carling speculates in his Short comment about the unconstrained contribution of older glaciations, preceding the last glacial maximum, to the fluvial transport distances estimated in our study. Referee #2 askes for paleo-ice limits indicated in Figure 1. Data on previous glacial extend prior to the last glacial maximum is scarce on the central Himalayan front (e.g. Owen, 2020). We decided to indicate the evidence of U-shaped valley topography for maximum Quaternary ice extend in the Langtang valley, observed by Shiraiwa and Watanabe (1991), to Figure 1. We hope that Figure 1 can now better illustrate (with this single paleo-ice limit) that glacial transport can only play a negligible role in transporting the surveyed boulders down the valley in the studied catchments

Shiraiwa, T. and Watanabe, T., 1991. Late Quaternary glacial fluctuations in the Langtang valley, Nepal Himalaya, reconstructed by relative dating methods. Arctic and Alpine Research, 23(4): 404-416.

[Figure]

**Commented [A39]:** Following a suggestion by Referee #2: White boxes were put behind exposure ages for better readability.

**Figure 2:** A- Boulders lying on a tributary fan south of Devighat, Trishuli valley. The main valley widens substantially at this location. Sample from top to bottom (as shown in the Figure): NEQ/162 47, ...46, ...45, ...44. B- Boulder appears sub-angular sitting on top of terrace deposit at Betrawatti (NEQ/162 66), Trishuli valley. In the background peaks rise >5000 m. C- Narrow part of Sunkoshi river after its confluence with Balephi Khola. Sample from top to bottom (as shown in the Figure): NEQ/161 02, ...01. D- Largest boulder surveyed in this study (NEQ/161 03) in Sunkoshi valley with an intermediate diameter of 29.9 m consists of gneiss lithology (Ulleri Gneiss), minimum travel distance 13 km. [10]Be surface exposure ages in ka BP. Coordinates of viewpoints given.

[Figure]

**Commented [ML40]:** Figure was updated with documented flow characteristics that induced boulder mobilisation during the 1981 and 2016 Sunkoshi GLOFs

**Commented [A41]:** Modifications made by the authors: it is not really an exemplary cross-section (except gradient), rather exemplary input parameters.

[revised manuscript text omitted]

---

## Author Response (AR2)

Marius Huber

Centre de Recherches Pétrographiques et Géochimiques

15 rue Notre Dame des Pauvres BP 20

54500 Vandœuvre les Nancy, France

Dr. Wolfgang Schwanghart

Institut für Umweltwissenschaften und Geographie

Karl-Liebknecht-Str. 24-25

14476 Potsdam-Golm, Germany

**Revised submission after review by editor and additional referee**

| | |
|---|---|
| Scientific journal: | Earth Surface Dynamics (ESurf), EGU Publications |
| Contribution title: | Timing of exotic, far-travelled boulder emplacement and paleo-outburst flooding in the central Himalaya |
| Authors: | Marius L. Huber, Maarten Lupker, Sean F. Gallen, Marcus Christl, and Ananta P. Gajurel |
| Received: | 28 Feb 2020 |
| Revised submission: | 21 Jun 2020 |

Dear Dr Schwanghart,

Valuable comments by you and an additional Anonymous Referee are treated in this letter by a point-by-point answer below:

- **Associate Editor Decision, comments for minor revisions**
- **Referee comment by Anonymous Referee, Referee #3**

Following yours and the additional Referee's suggestions we mainly implemented additional references which further increase the quality of our contribution in this late stage of manuscript preparation. Besides this improvement of the bibliography cited we added only minor modifications to the text where we thought it was appropriate.

We would like to thank you and the Referee #3 for constructive additional feedback. The e-surf editorial board shall be thanked again for the efficient handling.

Marius Huber, on behalf of all the co-authors.

Reviewer's comment

*Answer to comment*

Modification in the manuscript
* * *
**Associate Editor Decision, comments for minor revisions (W. Schwanghart)**

[1] L391f: The analysis of recorded meteorological floods could be potentially complemented by referencing the paper by Rakhecha (2002, Rakhecha, P. R.: Highest floods in India. The Extremes of the Extrems: Extraordinary Floods. Proceedings of a symposium held at Reykjavik. Iceland, July 2000, pp. 167–172., 2002.), who present an envelope curve for extreme floods conditional on drainage area. I think that this paper likely strengthens your argument that present day meteorological floods are likely unable to have discharges necessary to transport the boulders analyzed in this study.

*The authors consulted the mentioned reference. After some discussion, we however, decided against incorporating it in this revised version of the manuscript. While this publication suggests the existence of large floods (based on a small but world-wide flood catalogue) that could reach discharges close to the paleo-discharge estimates we made for the emplacement of the studied boulders, we are mainly bothered by the small dataset and absence of information on the methodology. It is also unclear how these estimates could be transported to the central Himalaya, where large parts of the catchments are located on the dryer and higher elevation Tibetan plateau. We believe that by using the hydrological data from nearby stations (Figure 5) we offer a more robust estimate of the local flood hydrology compared to the broader compilation of Rakhecha (2002).*

[2] L423: A more recent publication that underscores the possibility of Mw 9 earthquakes is by Stevens and Avouac (2016, Stevens, V. L. and Avouac, J.-P.: Millenary Mw > 9.0 earthquakes required by geodetic strain in the Himalaya, Geophys. Res. Lett., 43(3), 2015GL067336, doi:10.1002/2015GL067336, 2016.)

*We added this more recent reference to the manuscript. Stevens and Avouac (2016) support the evidence that $M_W > 9.0$ earthquakes are very likely and have estimated recurrence intervals of >800 years at the whole Himalayan arc scale. Although it remains possible, we have mentioned in the text that we deem it as unlikely that LLOFs are solely related to Mw >9 earthquakes.*

**Section 5.4, p.14:** Larger events, such as the possibility for ≥ Mw 9.0 earthquakes, with a suggested recurrence interval of >800 years along the entire Himalayan arc (Stevens and Avouac, 2016), could also trigger rare and large LLOFs. However, large hillslope failures and valley fills have been shown to occur for lower earthquake magnitudes as well (e.g. Schwanghart et al., 2016a), and other controls on landslide initiation are likely important factors as well, e.g. hillslope saturation (Lu and Godt, 2013). We, therefore, consider it unlikely that LLOFs are restricted to (≥ Mw 9.0) events.

[3] Though listed in the discussion version, the reference below has not been in the final version:

Fort, M., Cossart, E., Arnaud-Fassetta, G., 2010. Hillslope-channel coupling in the Nepal Himalayas and 520 threat to man-made structures: The middle Kali Gandaki valley. Geomorphology 124, 178–199.

Considering the orientation of the final version, I think the reference could be re-incorporated in (1) chapter 5.4 because it is a good example of "minor" LLOF, yet with displacement of large boulders now acting as bed armouring, and/or (2) in chapter 5.5 because it also shows (i) the regulation of fluvial incision, and (ii) the incidence on the future on the road network.

*We agree with the relevance of this reference and added it in section 5.5 as suggested. Fort et al. 2010 discuss the persistence of large boulders in the channel and their potential temporal impact on river incision. For section 5.4, in which we discuss potential triggers of Holocene catastrophic LOFs, we could not find a suitable location to add a reference for a minor LLOF to the text and have therefore not changed the text.*

**Section 5.5, p.15:** Large grain-size boulders in channel beds also have the potential to affect long-term channel incision patterns (e.g. Fort et al., 2010; Shobe et al., 2016; 2018).
* * *
**Referee #3 (Anonymous)**

[5] Line 403, the authors refer to climate variability, but they refer only to "modulation of glacier extent and proglacial lake volumes". Referring on recent LLOFs following monsoon rainfall events could also be relevant (i.e. Tatopani (1998) in the Kali Gandaki as a good example), even if they then consider these LLOF cannot provide discharge high enough to move such large boulders.

*Climate variability can also affect monsoonal rainfall pattern and intensity. We acknowledge that we omitted a meteorological trigger in the sentence mentioned. But we would like to argue that meteorologically triggered floods, causing outburst or not, do unlikely affect two neighbouring catchments of the respective sizes at the same time with similar strength. We modified the sentence mentioned by Referee #3, as suggested by him/her.*

**Section 5.4, p. 13:** Climate variability can directly affect precipitation patterns and intensity during the monsoon or indirectly affect glacial dynamics, through its modulation of glacier extent and proglacial lake volumes and can therefore also be invoked as a potential trigger of large LOF events in multiple valleys during a short period of time, as observed in our data.

**Section 5.4, p.14:** Climate change may be another LOF trigger. Given the large size of the studied catchments as well as the modern discharge record (Figure 5), it is unlikely that exceptional and localised extreme rain fall could occur synchronously in two large valleys. However, climate through its modulation of glacier dynamics and the creation of proglacial lakes could control the occurrence of GLOFs.

[6] Line 663: Why not adding roads and related infrastructure (i.e. bridges) to hydropower? This is a great concern for a country like Nepal where road development is progressing very rapidly. See the paper (this is just one example):

McAdoo B.G., Quak M., Gnyawali K.R., Adhikari B.R., Devkota S., Rajbhandari P.L., Sudmeier-Rieux K., 2018. Roads and landslides in Nepal: How development affects environmental risk. Natural Hazards and Earth Systems Sciences, 18, 3203-3210. DOI: 10.5194/nhess-18-3203-2018

*This is a good idea! Thank you, Referee #3 for pointing towards it. We added 'road development' to a sentence in section 5.5 addressing GLOF hazard and risk, next to 'hydropower', as Referee #3 suggested. We cited the publication proposed by Referee #3 behind this sentence as an example (McAdoo et al., 2018).*

[revised manuscript text omitted]